# Flow Matching for Few-Trial Neural Adaptation with Stable Latent Dynamics

**Puli Wang** [1 2]  **Yu Qi** [2 3]  **Yueming Wang** [1 2]  **Gang Pan** [1 2]

## Abstract

The primary goal of brain-computer interfaces (BCIs) is to establish a direct linkage between neural activities and behavioral actions via neural decoders. Due to the nonstationary property of neural signals, BCIs trained on one day usually obtain degraded performance on other days, hindering the user experience. Existing studies attempted to address this problem by aligning neural signals across different days. However, these neural adaptation methods may exhibit instability and poor performance when only a few trials are available for alignment, limiting their practicality in real-world BCI deployment. To achieve efficient and stable neural adaptation with few trials, we propose Flow-Based Distribution Alignment (FDA), a novel framework that utilizes flow matching to learn flexible neural representations with stable latent dynamics, thereby facilitating source-free domain alignment through likelihood maximization. The latent dynamics of FDA framework is theoretically proven to be stable using Lyapunov exponents, allowing for robust adaptation. Further experiments across multiple motor cortex datasets demonstrate the superior performance of FDA, achieving reliable results with fewer than five trials. Our FDA approach offers a novel and efficient solution for few-trial neural data adaptation, offering significant potential for improving the long-term viability of real-world BCI applications.

[1]College of Computer Science and Technology, Zhejiang University [2]The State Key Lab of Brain-Machine Intelligence, Zhejiang University [3]MOE Frontier Science Center for Brain Science and Brain-Machine Integration, Zhejiang University. Correspondence to: Yu Qi <qiyu@zju.edu.cn>, Gang Pan <gpan@zju.edu.cn>.

*Proceedings of the 42nd International Conference on Machine Learning*, Vancouver, Canada. PMLR 267, 2025. Copyright 2025 by the author(s).

## 1. Introduction

The aim of Brain-computer Interfaces (BCIs) is to establish a direct link between the brain and external devices, presenting great opportunities for improving neural rehabilitation in individuals with paralysis (Willett et al., 2021; 2023; Wu et al., 2016; Wang et al., 2023a). However, sustaining long-term decoding performance in chronic implantation is challenging, resulting from behavioral variability (Truccolo et al., 2008), physiological changes (Athalye et al., 2017), and device degradation (Woeppel et al., 2021). The dynamic relationship between neural data and behavior necessitates recalibrating neural representations through adaptation to ensure high-performance behavioral decoding.

Existing work on neural adaptation (Dabagia et al., 2023) have focused on latent embeddings and aligned them for stable long-term neural decoding. For example, linear methods, such as principal component analysis (PCA) (Degenhart et al., 2020), are used for aligning interpretable latent factors. Non-linear methods based on low-dimensional latent spaces usually have explicit assumptions on the statistical properties of latent variables. NoMAD (Karpowicz et al., 2022) and the source-free alignment (Vermani et al., 2024) based on seq-VAEs assume Gaussian posteriors for the closed-form distribution divergences.

However, these methods may lack stability and efficiency when only a few trials are available for alignment in real-world BCI deployment, which requires minimal recalibration (Karpowicz et al., 2024) over extended periods. For instance, empirical results for two typical methods are shown in Fig. 1(a). While they achieved reasonable performance with a substantial number of target trials ($\sim 100$), they failed to maintain effective behavioral decoding with fewer than 5 target trials. This issue may be attributed to two main factors. First, some alignment techniques assume prior distributions for latent variables to simplify likelihood estimation (Karpowicz et al., 2022; Wang et al., 2023b; Vermani et al., 2024). Nonetheless, this assumption may be invalid when target trials are limited. NoMAD relies on the closed-form KL-divergence of Gaussian distributions, which may lead to negative transfer (Pan & Yang, 2009) in few-trial scenarios, as shown in Fig. 1(b)(Left). Second, training of certain alignment methods, such as Cycle-GAN (Ma et al., 2023), might be quite unstable with few trials as illustrated

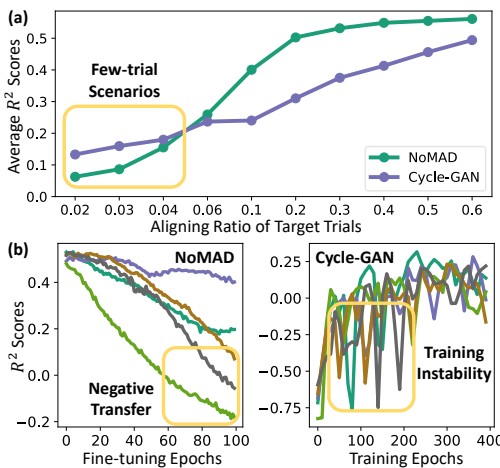

*Figure 1.* (a) Average $R^2$ across target sessions for NoMAD and Cycle-GAN with varying numbers of target trials. (b) Representative $R^2$ curves of NoMAD and Cycle-GAN on test target trials, with an aligning ratio of 0.02.

in Fig. 1(b)(Right), due to the discontinuity of divergence functions (Arjovsky et al., 2017).

This necessitates methods that enable few-trial adaptation to achieve stable neural representations, thereby sustaining high-performance behavioral decoding over time. To achieve efficient few-trial neural adaptation, we propose a novel Flow-Based Distribution Alignment (FDA) framework that leverages recent advances in flow matching (Lipman et al., 2023). The FDA method learns stable neural representations with flexible distributions rather than prior stationary distributions. Our flow-based learning also enables a source-free alignment through direct likelihood maximization that is less sensitive to few-trial scenarios (Wang et al., 2023b). The latent dynamics of our flow-based transformations are theoretically proven to be stable, based on Lyapunov exponents. Extensive experiments on multiple motor cortex datasets confirm the superior performance of our FDA, achieving a 15% improvement in $R^2$ over existing methods with fewer than 5 trials. Our FDA approach establishes an innovative framework for efficient and robust adaptation in few-trial scenarios, potentially improving the long-term reliability of real-world BCIs (Dabagia et al., 2023; Fan et al., 2023; Karpowicz et al., 2024). The main contributions of this paper are summarized as follows:

- **Flow-Based Distribution Alignment (FDA)**: Flow matching was first employed to enable efficient few-trial neural adaptation. Our proposed Flow-Based Distribution Alignment (FDA) framework learns stable neural representations, allowing for source-free alignment through likelihood maximization.

- **Stable Latent Dynamics**: The FDA method is demonstrated to be stable using Lyapunov exponents. This stability is evident in the tendency of neural representations to remain within an invariant bounded set (Kolter

& Manek, 2019), enabling robust adaptation.

- **Experimental Validation**: We extensively validated our FDA method on several motor cortex datasets (Ma et al., 2023), achieving reliable decoding with fewer than five target trials.

## 2. Related Work

**Neural Representation Alignment** To address the challenge of variability in neural recordings, several approaches (Duan et al., 2023a;b; Qi et al., 2019; Zhu et al., 2022) have been proposed to enable robust behavioral decoding, including unsupervised alignment of neural representations (Dabagia et al., 2023). For example, ADAN (Farshchian et al., 2018) and Cycle-GAN (Ma et al., 2023) achieved neural alignment by employing adversarial learning techniques based on raw neural signals. Latent features such as low-dimensional neural manifolds (Jude et al., 2022; Karpowicz et al., 2022; Wang et al., 2023b; Vermani et al., 2024) were aligned across sessions. As an example, ERDiff (Wang et al., 2023b) uses diffusion models and aligns the target distribution with that extracted from VAEs. In addition, transformer-based models (Liu et al., 2022; Ye et al., 2023; Azabou et al., 2023; Zhang et al., 2024; Wang et al., 2025) can achieve supervised neural alignment on target downstream tasks via fine-tuning, following pre-training with self-supervised objectives such as masked reconstruction. In contrast, our FDA achieves representation learning entirely through diffusion models, with source-free alignment based solely on direct log-likelihood maximization. In cases with few target trials, alignment approaches that rely on one-to-one sample mapping tend to fall into suboptimal solutions (Courty et al., 2017; Kerdoncuff et al., 2021). Unlike these methods, likelihood-based alignment is less sensitive to the number of target trials (Wang et al., 2023b). Consequently, the FDA achieves efficient alignment in few-trial scenarios, a challenge that most existing unsupervised alignment has not effectively addressed.

**Normalizing Flows** Normalizing flows have been widely applied in distribution sampling due to their explicit likelihood modeling. For example, flow matching (Liu et al., 2023a; Lipman et al., 2023; Ma et al., 2024) extends diffusion models with continuous normalizing flows for more flexible diffusion paths. Conditional flow matching (Liu et al., 2023b) further integrates conditional features to model conditional distributions. Additional related work is provided in App. A.4. Unlike those typically used for learning neural dynamics (Kim et al., 2021), these normalizing flows are employed for distribution sampling, where only the final latent state is taken as the sample. Existing alignment methods (Gong et al., 2019; Liu et al., 2023a) based on normalizing flows typically start from the source distribution and end with the target one. Based on conditional flow matching, we propose a more efficient strategy in few-trial

scenarios, which have not been explored in the context of neural alignment.

# 3. Methodology

## 3.1. Problem Formulation

We define the problem of long-term behavioral decoding in few-trial scenarios based on the unsupervised domain adaptation (Long et al., 2013). First, we define the domain $\mathcal{D} = \{(x_1, y_1), \ldots, (x_n, y_n)\}$, where $x_i(l)(l = 1, 2, \ldots, m)$ represents the raw neural signal sample from the $l$-th channel in one or more sessions. The short-term context window has a length of $w$ time points, much smaller than the length of trials, i.e., $x_i(l) \in \mathbb{R}^w$. The first context window of each trial begins at the initial time point, while the second window starts one step later. The temporal evolution of neural dynamics is reflected in the shifting of short-term windows. $y_i$ denotes the $d$-dimensional behavioral label corresponding to the $w$-th time step of $x_i$, with $y_i \in \mathbb{R}^d$. The behavioral label is assigned at the $w$-th time step to leverage previous time steps as contextual information.

Based on $\mathcal{D}$, we define the source domain $\mathcal{D}_S$, consisting of both signals and labels from one or more sessions: $\mathcal{D}_S = \{(x_1^S, y_1^S), \ldots, (x_{n_S}^S, y_{n_S}^S)\}$. Similarly, the unlabeled target domain $\mathcal{D}_T$ consists of signals from a separate session: $\mathcal{D}_T = \{x_1^T, \ldots, x_{n_T}^T\}$, where $n_T \ll n_S$, and typically only contains signals of few trials. For convenience, we define $\mathbf{x}^S$ and $\mathbf{y}^S$ as the random variables representing neural signals $x_i^S$ and their corresponding labels $y_i^S$ in $\mathcal{D}_S$. Samples $x_j^T$ from $\mathcal{D}_T$ are represented as random variables $\mathbf{x}^T$. We aim to align the distribution of neural representations from $\mathcal{D}_T$ with $\mathcal{D}_S$ with few target trials, reusing behavioral decoders trained on source behavioral labels $\mathbf{y}^S$.

## 3.2. Overall framework

To obtain efficient and robust adaptation in few-trial scenarios, we propose a novel FDA framework that employs flow matching with stable latent dynamics and achieves source-free alignment through likelihood maximization. This framework consists of two phases: pre-training and fine-tuning, as illustrated in Fig. 2.

During the pre-training phase, we establish a continuous normalizing flow conditioned on context windows using $\mathcal{D}_S$ in a supervised manner. Initial noisy latent states flow toward the target neural representation for decoding, guided by short-term context windows from raw neural signals. There are no prior assumptions on the distribution of neural representations. The latent dynamics of our FDA are further verified to be stable. This stability is ensured by the Lipschitz continuity of activation functions in MLPs and by regularizing the drift coefficients of latent states, as detailed in Theorem 3.1.

As for fine-tuning, we perform unsupervised alignment of neural representation using few trials from $\mathcal{D}_T$. Compared to some existing flow-based adaptation methods (Gong et al., 2019; Liu et al., 2023a), FDA allows for alignment with fewer target trials. Based on the explicit computation of log-likelihood using the Fokker-Planck Equation, we propose a novel source-free alignment method through direct likelihood maximization.

### 3.2.1. FLOW-BASED REPRESENTATION LEARNING

During the pre-training phase, we propose a novel framework based on flow matching conditioned on context windows. FDA offers several distinct benefits in obtaining neural representations. First, flow matching imposes fewer assumptions on the underlying statistics of latent variables, enabling more flexible modeling of distributions and resolving the breakdown of prior assumptions in few-trial scenarios. Second, our theoretical analysis demonstrates that the latent dynamics of our flow-based transformations are stable (Angeli, 2002), which is further validated using Lyapunov exponents. This stability is manifested in the tendency of neural representations to stay within an invariant bounded set (Kolter & Manek, 2019), which allows for efficient and robust alignment for few trials.

**Conditional Feature Extraction** We begin by extracting conditional features $c_i^S$ from context windows $x_i^S$. For spike-based signals, a single channel usually records neuron-level activity (Buzsáki, 2004), where the short-term patterns are relatively limited for similar tasks (Izhikevich et al., 2004). Moreover, inter-channel relationships in spikes are generally more stable compared to the temporal patterns, which often exhibit warping (Williams et al., 2020). The above observations are validated, as demonstrated in Fig. 5(c).

Based on these observations and inspired by (Liu et al., 2024), we use short-term temporal patterns of each channel as tokens, along with their inter-channel relationships for attention calculation. This approach can flexibly accommodate changes in the number of channels, which is quite common during neuron growth and apoptosis (Degenhart et al., 2020). Specifically, we feed the raw neural signal sequence $x_i^S = [x_i^S(1), \ldots, x_i^S(m)]$, containing tokens from $m$ channels, into a transformer-based network $f_\alpha$ (with parameters $\alpha$) using the classical sinusoidal positional encoding. After processing through multi-head self-attention modules and projection networks, we obtain conditional latent dynamics: $c_i^S = f_\alpha(x_i^S)$, where $c_i^S \in \mathbb{R}^{k_c}$. The detailed architecture is illustrated in App. A.1.

**Flow Matching to Neural Representation** After extracting conditional features, we leverage flow matching conditioned on these features to learn the neural representations for decoding. We model the conditional probability $p_\tau(\mathbf{z}^S(\tau)|\mathbf{c}^S)$ using probability flow ODEs (Song et al., 2021), where

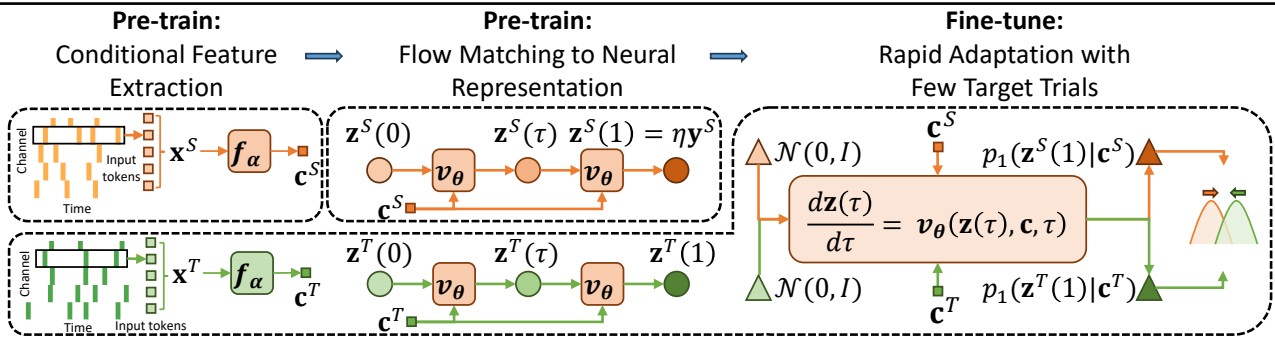

*Figure 2.* Two phases of the overall FDA framework: pre-training, which involves conditional feature extraction and flow matching to neural representation, and fine-tuning, which enables rapid adaptation with few target trials.

$\mathbf{z}^S(\tau) \in \mathbb{R}^{k_z}$ denotes the latent states at learning time point $\tau \in [0, 1]$, where $\tau$ is rescaled for more efficient training, capturing the evolution of $\mathbf{z}^S$. Here, $\mathbf{c}^S$ is the random variable representing conditional features $c_i^S$ ($\mathbf{c}^S = f_\alpha(\mathbf{x}^S)$). Typically, the flows are built on the parameterized $\phi_\tau$ to transform a simple prior distribution $p_0$ (e.g., a multivariate Gaussian) into a more complex one $p_1$: $p_\tau = [\phi_\tau]_* p_0$.

For adaptation to the supervised training on $\mathcal{D}_S$, we set $p_0$ as a standard multivariate Gaussian distribution, i.e., $\mathbf{z}^S(0) \sim \mathcal{N}(0, I)$. The target distribution $p_1$, representing the desired neural representation for behavioral decoding, is defined by the random variable $\mathbf{z}^S(1) = \eta \mathbf{y}^S$, where $\eta \in \mathbb{R}^{k_z \times d}$ is pre-defined with Xavier initialization and remains the same across sessions. This distribution is denoted as $q(\mathbf{z}^S(1))$, with $\eta^* \in \mathbb{R}^{d \times k_z}$ as the generalized inverse of $\eta$, which serves as weights of the linear decoder $G$ and also satisfies $\eta^* \eta = I_d$. In the detailed implementation, the flow $\phi_\tau$ of $p_\tau(\mathbf{z}^S(\tau) | \mathbf{c}^S)$ is optimized following the objectives used by conditional flow matching. A neural network $v_\theta$ (with parameters $\theta$) is utilized to parameterize the vector field of $\mathbf{z}^S(\tau)$, allowing for its evolution as follows:

$$\frac{d\mathbf{z}^S(\tau)}{d\tau} = v_\theta(\mathbf{z}^S(\tau), f_\alpha(\mathbf{x}^S), \tau). \quad (1)$$

Based on Eq. (1), the evolution of $p_\tau(\mathbf{z}^S(\tau) | \mathbf{c}^S)$ during learning process follows the Fokker-Planck Equation:

$$\frac{\partial p_\tau(\mathbf{z}^S(\tau) | \mathbf{c}^S)}{\partial \tau} = -\nabla \cdot \left( p_\tau(\mathbf{z}^S(\tau) | \mathbf{c}^S) \, v_\theta(\mathbf{z}^S(\tau), f_\alpha(\mathbf{x}^S), \tau) \right). \quad (2)$$

Existing work (Liu et al., 2023b) indicates that the network $v_\theta$ can be optimized via matching the vector field provided by $v_\theta$ with a predefined vector field $u(\tau)$. To enhance the efficiency in learning and distribution alignment of neural representation, we set the flow path over our learning process as a linear interpolation between the start $\mathbf{z}^S(0)$ and the end $\mathbf{z}^S(1)$:

$$\mathbf{z}^S(\tau) = (1 - \tau)\mathbf{z}^S(0) + \tau \mathbf{z}^S(1). \quad (3)$$

The corresponding vector field of Eq. (3) is $u(\tau) = \mathbf{z}^S(1) - \mathbf{z}^S(0)$. Based on these, the training objective function

$\mathcal{L}_{\mathrm{cfm}}(\alpha, \theta)$ can be defined as below:

$$\mathcal{L}_{\mathrm{cfm}}(\alpha, \theta) = \mathbb{E}_{\tau, p(\mathbf{z}^S(0)), q(\mathbf{z}^S(1))}$$
$$\left\| v_\theta(\mathbf{z}^S(\tau), f_\alpha(\mathbf{x}^S), \tau) - (\mathbf{z}^S(1) - \mathbf{z}^S(0)) \right\|, \quad (4)$$

where $\mathbf{z}^S(0) \sim \mathcal{N}(0, I)$, $\mathbf{z}^S(1) = \eta \mathbf{y}^S$, and $\|\cdot\|$ denotes the $\ell_2$ norm. $v_\theta$ only consists of multilayer perceptron (MLP) layers with residual connections, and its detailed architecture is provided in App. A.1. More intuitively, we provided a visualization of the flow matching process in Fig. 3. Our FDA learns latent variables from neural signals in a coarse-to-fine manner, differing from conventional one-step extraction. The stable latent dynamics of the learning process ensure that latent factors with similar labels flow toward consistent neural representations, even when guided by shifted neural recordings. Empirical validation through zero-shot transfer performance in Table S8 further demonstrates the benefits of this stability.

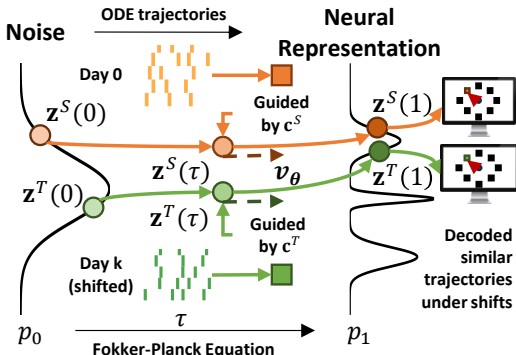

*Figure 3.* Intuitive visualization of our flow matching process. Our process transforms noisy variables $\mathbf{z}(0)$ into an ideal neural representation $\mathbf{z}(1)$, guided by conditional features $\mathbf{c}$ derived from neural signals. Meanwhile, the corresponding distribution $p_0$ is transformed into $p_1$. To learn this process, we parameterize the velocity field $v_\theta$ of $\mathbf{z}$, which determines the moving velocity and direction of $\mathbf{z}$ at a specific learning time point $\tau$.

**Theoretical Analysis on Latent Dynamics** Our flow-based neural representation learning forms a dynamic process, beginning with noisy latent states and ending with the desired

neural representation. We further found that the latent dynamics of this transformation process are stable (Angeli, 2002), which is ensured by two key factors. First, the velocity field in flow matching is constructed using MLPs with Lipschitz-continuous activation functions. These functions ensure that latent state deviations remain stable under external input constraints, as shown in Eq. (7) and Eq. (21). Second, the scale coefficient $\gamma^S$, which controls the shift of $\mathbf{z}$ in predicting the velocity field as defined in Eq. (14), is regularized to ensure that the ratio of latent state deviations between successive time steps remains below 1. This results in a geometric sequence with a ratio less than 1, causing latent states to gradually converge to similar ones, as presented in Eq. (6) and Eq. (22).

Consider any two signal samples $x_i^S$ and $x_j^S$ from $\mathcal{D}_S$, with corresponding conditional features $c_i^S$ and $c_j^S$, and their latent states $\mathbf{z}_i^S(\tau)$ and $\mathbf{z}_j^S(\tau)$. We then analyze the upper bound of the distance $\left\| \mathbf{z}_i^S(\tau) - \mathbf{z}_j^S(\tau) \right\|$ based on the Euler sampling method. We summarize the verification in Theorem 3.1 below. Detailed proof can be found in App. A.2.

**Theorem 3.1.** *Let the total number of sampling steps in Euler's method be $\mathcal{T}$. At the $n$-th step, the time point is $\tau_n = \frac{n}{\mathcal{T}}$. At this point, the distance between any two latent states $z_i^S(\tau_n)$ and $z_j^S(\tau_n)$ corresponding to signal samples $x_i^S$ and $x_j^S$ satisfies the following inequality:*

$$\|z_i^S(\tau_n) - z_j^S(\tau_n)\| \leq$$
$$h_z\left(\|z_i^S(0) - z_j^S(0)\|, n\right) + h_c\left(\|c_i^S - c_j^S\|\right), \quad (5)$$

*where $h_z : \mathbb{R}_{\geq 0} \times \mathbb{Z}_{\geq 0} \to \mathbb{R}_{\geq 0}$ is a decreasing function with respect to $n$, given by:*

$$h_z\left(\|z_i^S(0) - z_j^S(0)\|, n\right) = (\mathbf{K}_\gamma)^n \|z_i^S(0) - z_j^S(0)\|, \quad (6)$$

*with $0 < \mathbf{K}_\gamma < 1$. Moreover, $h_c : \mathbb{R}_{\geq 0} \to \mathbb{R}_{\geq 0}$ satisfies $h_c\left(\|c_i^S - c_j^S\|\right) \to \infty$ as $\|c_i^S - c_j^S\| \to \infty$. The function $h_c\left(\|c_i^S - c_j^S\|\right)$ can be expressed as:*

$$h_c\left(\|c_i^S - c_j^S\|\right) = \left(\sum_{a=1}^{n-1} (\mathbf{K}_\gamma)^a\right) \mathbf{K}_g \|\mathbf{w}_\beta\| \|c_i^S - c_j^S\|, \quad (7)$$

*where $\mathbf{K}_g$ is the Lipschitz constant of activation functions in the network $v_\theta$, and $\mathbf{w}_\beta$ represents the weights used for computing shift coefficients (Ma et al., 2024) in $v_\theta$.*

### 3.2.2. RAPID ADAPTATION WITH FEW TARGET TRIALS

During the fine-tuning phase, the pre-trained flow network $v_\theta$ is fixed, while the conditional feature extractor $f_\alpha$ is fine-tuned, aligning the distribution of neural representation $\mathbf{z}(1)$. The flow path is approximated as a straight line, allowing us to obtain the final latent states in a single step for decoding. This significantly simplifies the explicit computation of likelihood functions, which ERDiff (Wang et al.,

2023b) finds challenging. Therefore, unlike ERDiff, which maximize the lower bound of log-likelihood, we propose a direct log-likelihood maximization approach that achieves source-free unsupervised alignment with few trials.

**Source-Free Alignment via Likelihood Maximization (FDA-MLA)** A notable advantage of flow matching is its explicit modeling of likelihood. Meanwhile, distribution alignment based on minimizing Kullback–Leibler (KL) divergences can be seen as maximizing the likelihood in $\mathcal{D}_T$ (Kingma et al., 2019). Moreover, our FDA does not directly rely on source samples, making it suitable for privacy-sensitive data like neural recordings, enabling source-free unsupervised alignment.

Specifically, let the signal samples in $\mathcal{D}_T$ be denoted by the random variable $\mathbf{x}^T$, with the corresponding conditional feature $\mathbf{c}^T = f_\alpha(\mathbf{x}^T)$ and the latent embedding $\mathbf{z}^T(1)$ for decoding. In this context, aligning the final latent state of flow between $\mathcal{D}_S$ and $\mathcal{D}_T$ can be achieved by minimizing the KL divergence. This can be accomplished by fine-tuning the parameters $\alpha$ of conditional feature extractor $f_\alpha$:

$$\min_\alpha D_{\mathrm{KL}}\left(p_1(\mathbf{z}^S(1)|f_\alpha(\mathbf{x}^S)) \parallel p_1(\mathbf{z}^T(1)|f_\alpha(\mathbf{x}^T))\right)$$
$$\approx \max_\alpha \log p_1(\mathbf{z}^T(1)|f_\alpha(\mathbf{x}^T)). \quad (8)$$

Since minimizing KL divergences can be approximated as maximizing log-likelihood functions, we can reformulate the above objective function as maximizing the likelihood based on $p_1(\mathbf{z}^T(1)|f_\alpha(\mathbf{x}^T))$, which reduces dependence on $\mathcal{D}_S$. Furthermore, our pre-defined flow path is approximated as a straight line, the neural representation can be sampled using the one-step Euler method. This also simplifies the computation of likelihood functions for target conditional probabilities.

Taking one-step Euler sampling as an example, the likelihood of this conditional probability can be explicitly expressed via the change of variables formula (Chen et al., 2018) as:

$$\log p_1(\mathbf{z}^T(1)|f_\alpha(\mathbf{x}^T)) = \log p_0(\mathbf{z}^T(0)|f_\alpha(\mathbf{x}^T))$$
$$- \log\left|\det\left(\frac{\partial v_\theta(\mathbf{z}^T(0), 0, f_\alpha(\mathbf{x}^T))}{\partial \mathbf{z}^T(0)}\right)\right|. \quad (9)$$

Considering that $\log p_0(\mathbf{z}^T(0)|f_\alpha(\mathbf{x}^T))$ is independent of $\alpha$, the objective function $\max_\alpha \mathcal{L}_{\mathrm{mla}}(\alpha)$ can be approximately rewritten as below through target neural signals $x_j^T$:

$$\max_\alpha \left(\sum_{j=1}^{n_T} -\log\left|\det\left(\frac{\partial v_\theta(z_j^T(0), 0, f_\alpha(x_j^T))}{\partial z_j^T(0)}\right)\right|\right), \quad (10)$$

where $z_j^T(1) = v_\theta(z_j^T(0), 0, f_\alpha(x_j^T))$.

More generally, alternative sampling methods can employ the unbiased Hutchinson-trace estimator (Hutchinson, 1989) to estimate the divergence in Eq. (2), facilitating effective alignment through likelihood maximization. Detailed computations are provided in App. A.3.

**Maximum Mean Discrepancy Alignment (FDA-MMD)** For a fair comparison with alignment methods using source data, we also employ a strategy that minimizes MMD (Maximum Mean Discrepancy) distances to align the representation distributions of our flow-based learning. Taking one-step Euler sampling as an example, the objective function $\min_\alpha \mathcal{L}_{\mathrm{mmd}}(\alpha)$ for aligning final states $\mathbf{z}(1)$ based on $\mathcal{D}_T$ is as follows:

$$\min_{\alpha} \left\| \frac{1}{n_S} \sum_{i=1}^{n_S} \varphi(z_i^S(1)) - \frac{1}{n_T} \sum_{j=1}^{n_T} \varphi(z_j^T(1)) \right\|_{\mathcal{H}}, \quad (11)$$

where $z_i^S(1) = v_\theta(z_i^S(0), 0, f_\alpha(x_i^S))$, and $z_j^T(1) = v_\theta(z_j^T(0), 0, f_\alpha(x_j^T))$. Here, $\mathcal{H}$ represents the reproducing kernel Hilbert space (RKHS), and $\varphi$ is the feature mapping function in that space, and we utilize a Gaussian kernel to compute the inner product of features.

---

**Algorithm 1** Flow-Based Dynamical Alignment (FDA)

---
1: **Input:** source domain $\mathcal{D}_S$; target domain $\mathcal{D}_T$; alignment method $align\_m$; pre-defined $\eta$;
2: **Output:** conditional feature extractor $f_\alpha$; continuous normalizing flow network $v_\theta$
3: Initialize $f_\alpha$, $v_\theta$
4: **Pre-training phase:**
5: **for** $iter = 1$ **to** $n_{pre-train}$ **do**
6:     Sample $\tau$, $\mathbf{z}^S(0) \sim \mathcal{N}(0, I)$, $\mathbf{x}^S$, $\mathbf{z}^S(1) = \eta\mathbf{y}^S$;
7:     Update $f_\alpha$, $v_\theta$ by $\mathcal{L}_{\mathrm{cfm}}(\alpha, \theta)$;
8: **end for**
9: **Fine-tuning phase:**
10: **for** $iter = 1$ **to** $n_{fine-tune}$ **do**
11:     **if** $align\_m$ is FDA-MMD: **then**
12:         Sample $\mathbf{x}^S$, $\mathbf{z}^S(0) \sim \mathcal{N}(0, I)$ and $\mathbf{x}^T$, $\mathbf{z}^T(0) \sim \mathcal{N}(0, I)$; Update $f_\alpha$ by $\mathcal{L}_{\mathrm{mmd}}(\alpha)$;
13:     **else if** $align\_m$ is FDA-MLA: **then**
14:         Sample $\mathbf{x}^T$, $\mathbf{z}^T(0) \sim \mathcal{N}(0, I)$; Update $f_\alpha$ by $\mathcal{L}_{\mathrm{mla}}(\alpha)$;
15:     **end if**
16: **end for**
17: **return** $f_\alpha$, $v_\theta$.

---

### 3.3. Overall Learning Algorithm

The overall learning algorithm is illustrated in Algorithm 1. During the pre-training phase, we perform supervised optimization of the conditional feature extractor $f_\alpha$ and the flow network $v_\theta$ using $\mathcal{D}_S$, with the objective function $\mathcal{L}_{\mathrm{cfm}}(\alpha, \theta)$. In the fine-tuning phase, the parameter $\theta$ is fixed, and few trials from $\mathcal{D}_T$ are utilized to fine-tune $\alpha$ based on either $\mathcal{L}_{\mathrm{mmd}}(\alpha)$ or $\mathcal{L}_{\mathrm{mla}}(\alpha)$, as described in Sec. 3.2.2. Further training details are provided in App. B.2.

## 4. Experiments and Results

### 4.1. Experimental Setup

**Datasets** We employed three distinct datasets of extracellular neural recordings from the primary motor cortex (M1) of non-human primates (Ma et al., 2023). Additional information about the datasets can be found in App. B.1.
**Center-Out Reaching (CO-C&CO-M).** Monkeys C and M engaged in a center-out reaching task, where each trial required them to move to one of eight randomized targets, earning a reward for successful reaching.
**Random-Target (RT-M).** Monkey M performed a random-target task, reaching for three sequentially presented targets at random locations. Each trial started at the workspace center, with a 2.0-second limit to reach each target.
**Data Preprocess and Spilt** We extracted trials from the 'go cue time' to the 'trial end'. The data was then times-tamped and smoothed for firing rates in 50 ms bins. Sessions containing approximately 200 trials, along with 2D cursor velocity labels, were used as $\mathcal{D}_S$ for pre-training, while a separate session without labels was used as $\mathcal{D}_T$ for fine-tuning. For few-trial alignment, we used the target ratio $r$ to evaluate the number of target trials from all recorded ones, typically setting $r$ to 0.02, 0.03, 0.04, and 0.06, with 0.02 corresponding to no more than 5 trials. Considering the increased randomness in few-trial selections, we pre-train our FDA using 5 different random seeds and fine-tune it on 25 different random selections of few trials. The decoded cursor velocity is evaluated using $R^2$ scores, with results averaged over different selections and five pre-training processes. Additional experimental details and hyper-parameter settings can be found in App. B.2.

### 4.2. Comparative Study

**Baselines** The following approaches were utilized as baselines for comparative experiments, with further implementation details provided in App. B.3.
**LSTM**(Hochreiter, 1997): Unaligned LSTMs were used as baseline decoders to assess the challenges of alignment.
**CEBRA**(Schneider et al., 2023): CEBRA served as an advanced tool for discovering generalizable latent structures across datasets and subjects without alignment.
**ERDiff**(Wang et al., 2023b): ERDiff employed diffusion models to reconstruct spatio-temporal structures and aligned them with latent dynamics derived from VAEs.
**NoMAD**(Karpowicz et al., 2022): NoMAD utilizied LFADS (Pandarinath et al., 2018) to capture neural population dynamics and performed alignment.
**Cycle-GAN**(Ma et al., 2023): Cycle-GAN directly aligned full-dimensional raw signals at each time step through an adversarial approach.
Considering that NoMAD and ERDiff are both generative models, we included an additional reconstruction term in our

*Table 1.* Comparison of $R^2$ values (in %) of baselines and FDA on CO-M and RT-M datasets($r = 0.02$).

| Data | Session | LSTM | CEBRA | ERDiff | NoMAD | Cycle-GAN | **FDA-MLA** | **FDA-MMD** |
|---|---|---|---|---|---|---|---|---|
| CO-M | Day 0 | $74.18_{\pm4.90}$ | $79.24_{\pm1.38}$ | $82.71_{\pm2.82}$ | $79.77_{\pm4.50}$ | $77.06_{\pm2.21}$ | $\mathbf{84.79}_{\pm0.91}$ | $\mathbf{84.79}_{\pm0.91}$ |
| | Day 8 | $-118.53_{\pm98.70}$ | $-51.92_{\pm12.51}$ | $-0.14_{\pm60.88}$ | $15.32_{\pm11.96}$ | $14.25_{\pm10.29}$ | $23.79_{\pm8.71}$ | $\mathbf{45.23}_{\pm4.44}$ |
| | Day 14 | $-63.85_{\pm19.96}$ | $-1.77_{\pm7.03}$ | $-47.41_{\pm25.37}$ | $43.49_{\pm5.03}$ | $14.20_{\pm11.21}$ | $50.15_{\pm4.85}$ | $\mathbf{55.90}_{\pm3.17}$ |
| | Day 15 | $-712.91_{\pm316.04}$ | $-83.24_{\pm15.03}$ | $-49.80_{\pm19.89}$ | $20.26_{\pm7.52}$ | $9.77_{\pm6.36}$ | $43.59_{\pm3.69}$ | $\mathbf{49.55}_{\pm3.41}$ |
| | Day 22 | $-88.57_{\pm58.85}$ | $-21.10_{\pm7.01}$ | $-15.20_{\pm43.59}$ | $-7.71_{\pm39.52}$ | $14.10_{\pm5.22}$ | $\mathbf{33.98}_{\pm7.39}$ | $27.35_{\pm7.34}$ |
| | Day 24 | $-39.52_{\pm86.25}$ | $-10.28_{\pm3.35}$ | $-0.02_{\pm36.96}$ | $18.43_{\pm26.58}$ | $-3.14_{\pm14.96}$ | $48.86_{\pm4.58}$ | $\mathbf{51.28}_{\pm2.53}$ |
| | Day 25 | $-253.83_{\pm270.30}$ | $-64.67_{\pm16.20}$ | $-0.24_{\pm35.31}$ | $28.49_{\pm8.43}$ | $15.30_{\pm4.99}$ | $31.74_{\pm7.31}$ | $\mathbf{36.79}_{\pm4.12}$ |
| | Day 28 | $-107.64_{\pm124.47}$ | $-35.95_{\pm10.54}$ | $-9.79_{\pm40.68}$ | $32.68_{\pm3.98}$ | $0.35_{\pm14.38}$ | $53.27_{\pm7.55}$ | $\mathbf{54.87}_{\pm4.40}$ |
| | Day 29 | $-206.99_{\pm117.46}$ | $-64.32_{\pm15.75}$ | $-32.77_{\pm21.99}$ | $-37.13_{\pm69.37}$ | $16.32_{\pm2.99}$ | $36.16_{\pm9.21}$ | $\mathbf{41.26}_{\pm5.70}$ |
| | Day 31 | $-63.01_{\pm40.94}$ | $-81.41_{\pm21.04}$ | $-0.01_{\pm40.86}$ | $34.60_{\pm8.21}$ | $0.96_{\pm6.68}$ | $56.50_{\pm3.92}$ | $\mathbf{57.10}_{\pm3.24}$ |
| | Day 32 | $-417.39_{\pm295.63}$ | $-40.10_{\pm16.67}$ | $-7.58_{\pm34.99}$ | $28.52_{\pm4.68}$ | $6.18_{\pm13.31}$ | $40.49_{\pm5.69}$ | $\mathbf{44.66}_{\pm4.41}$ |
| RT-M | Day 0 | $72.91_{\pm1.40}$ | $74.86_{\pm1.03}$ | $76.98_{\pm2.62}$ | $74.71_{\pm2.87}$ | $85.19_{\pm2.36}$ | $\mathbf{86.95}_{\pm1.59}$ | $\mathbf{86.95}_{\pm1.59}$ |
| | Day 1 | $63.15_{\pm3.11}$ | $65.97_{\pm2.38}$ | $-0.62_{\pm0.05}$ | $58.72_{\pm19.32}$ | $32.38_{\pm2.33}$ | $71.83_{\pm3.90}$ | $\mathbf{74.32}_{\pm2.25}$ |
| | Day 38 | $-20.62_{\pm32.46}$ | $21.34_{\pm6.71}$ | $-18.38_{\pm25.62}$ | $25.87_{\pm10.12}$ | $21.55_{\pm3.36}$ | $55.05_{\pm2.65}$ | $\mathbf{55.39}_{\pm2.80}$ |
| | Day 39 | $-86.31_{\pm47.86}$ | $-36.86_{\pm25.62}$ | $-8.36_{\pm17.92}$ | $-6.25_{\pm11.77}$ | $-2.46_{\pm5.32}$ | $38.28_{\pm6.13}$ | $\mathbf{40.44}_{\pm7.31}$ |
| | Day 40 | $-8.36_{\pm17.70}$ | $2.63_{\pm20.16}$ | $-0.46_{\pm0.07}$ | $-3.51_{\pm12.01}$ | $22.02_{\pm11.65}$ | $32.16_{\pm8.95}$ | $\mathbf{39.85}_{\pm3.27}$ |
| | Day 52 | $3.12_{\pm11.68}$ | $30.50_{\pm6.94}$ | $-16.06_{\pm30.33}$ | $33.32_{\pm16.90}$ | $10.29_{\pm12.86}$ | $43.35_{\pm4.80}$ | $\mathbf{44.99}_{\pm4.96}$ |
| | Day 53 | $-43.50_{\pm50.26}$ | $42.33_{\pm4.84}$ | $-9.82_{\pm17.83}$ | $19.22_{\pm13.62}$ | $20.70_{\pm1.85}$ | $49.60_{\pm2.53}$ | $\mathbf{50.03}_{\pm4.44}$ |
| | Day 67 | $-148.64_{\pm98.52}$ | $25.09_{\pm13.79}$ | $-22.98_{\pm35.64}$ | $20.97_{\pm20.44}$ | $25.65_{\pm1.59}$ | $42.06_{\pm6.29}$ | $\mathbf{50.29}_{\pm1.07}$ |
| | Day 69 | $-110.99_{\pm93.95}$ | $-38.82_{\pm29.41}$ | $-0.33_{\pm0.02}$ | $-67.12_{\pm7.16}$ | $-5.99_{\pm27.79}$ | $29.52_{\pm7.31}$ | $\mathbf{39.19}_{\pm4.07}$ |
| | Day 77 | $-448.21_{\pm98.67}$ | $-53.79_{\pm21.04}$ | $-3.52_{\pm6.53}$ | $-31.65_{\pm16.18}$ | $-1.68_{\pm18.81}$ | $16.19_{\pm9.43}$ | $\mathbf{16.67}_{\pm9.32}$ |
| | Day 79 | $-226.00_{\pm135.06}$ | $-47.01_{\pm13.77}$ | $1.22_{\pm3.10}$ | $13.01_{\pm22.32}$ | $10.53_{\pm3.33}$ | $\mathbf{39.29}_{\pm6.86}$ | $38.99_{\pm5.70}$ |

method for a fair comparison and observed results similar to FDA without that regularization, as shown in Fig. 5(c)(FDA-re). Thus, we omitted the reconstruction term in the final version. Moreover, for fairness, the pre-training of NoMAD and ERDiff involved joint training of their generative models and behavioral decoders, while CEBRA was trained simultaneously on time and behavioral labels.

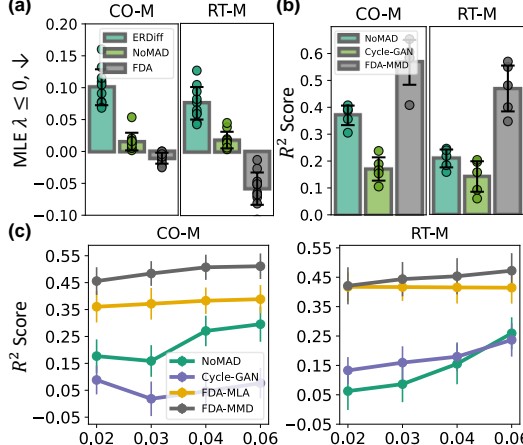

*Figure 4.* (a) The maximum Lyapunov exponent (MLE) $\lambda$ on CO-M and RT-M datasets. Dots represent the average MLE across five random runs of pre-training for each individual source session. Bar charts denote average MLE across sessions. (b) Comparison of $R^2$ scores for cross-session decoding ($r = 0.02$) with two sessions in $\mathcal{D}_S$. (c) Comparison of average $R^2$ scores across target sessions for baselines and FDA under varying $r$.

**Cross-session Performance Evaluation** We first validated the cross-session performance of FDA with few trials. We conducted experiments with $\mathcal{D}_S$ containing only one session. The average $R^2$ scores, using Day0 as the source session and a target ratio $r$ of 0.02, are presented in Table 1(CO-M & RT-M) and App. C.1.2(CO-C). FDA-MLA and FDA-MMD outperformed other methods across most sessions with few

trials. LSTM and CEBRA (without alignment) often yielded negative scores, emphasizing the need of alignment. Cycle-GAN and NoMAD performed much worse than reported in their original papers due to the scarcity of target trials, as shown in Fig. S5. ERDiff often experienced early stopping caused by gradient explosions in few-trial scenarios with its latest release, leading to performance degradation. Though FDA-MLA performed worse than FDA-MMD overall, this difference is understandable given that it is source-free.

We also performed comparisons with the two best baselines, NoMAD and Cycle-GAN. As shown in Fig. 4(c), FDA achieved a higher average $R^2$ across different values of $r$, demonstrating its efficient adaptation in few-trial scenarios. When $r$ increased to approximately 0.3 ($\sim$ 60 trials), FDA's performance became comparable to that of Cycle-GAN and NoMAD, as shown in Fig. S5. The overall performance of FDA across all sessions with few trials ($r = 0.02$) is presented in Fig. S3, demonstrating its consistently effective adaptation. Moreover, as presented in Fig. 4(b), FDA-MMD outperformed NoMAD and Cycle-GAN when $\mathcal{D}_S$ contained two sessions. We also found that FDA can achieve better adaptation in few-trial scenarios with more sessions in $\mathcal{D}_S$. Additional results can be found in App. C.1.2.

**Empirical Analysis on Latent Dynamics** To analyze the stability of latent dynamics in our flow-based transformations, we measured the maximum Lyapunov exponent (MLE) $\lambda$ of $\mathbf{z}^S(\tau)$ after pre-training on $\mathcal{D}_S$. Notice that the value of $\lambda$ was computed as described in (Wolf et al., 1985), with a non-positive $\lambda$ typically indicating dynamical stability. More detailed information on $\lambda$ are available in App. B.5. Since MLE operates on sequential variables, we compared our $\lambda$ with those derived from sequential latent factors (ERDiff & NoMAD). The results are presented in Fig. 4(a) and App. C.1.1. Nearly all MLEs achieved by FDA are non-positive (CO-M: 55/55, RT-M: 52/55), indicating greater stability compared to the selected baselines.

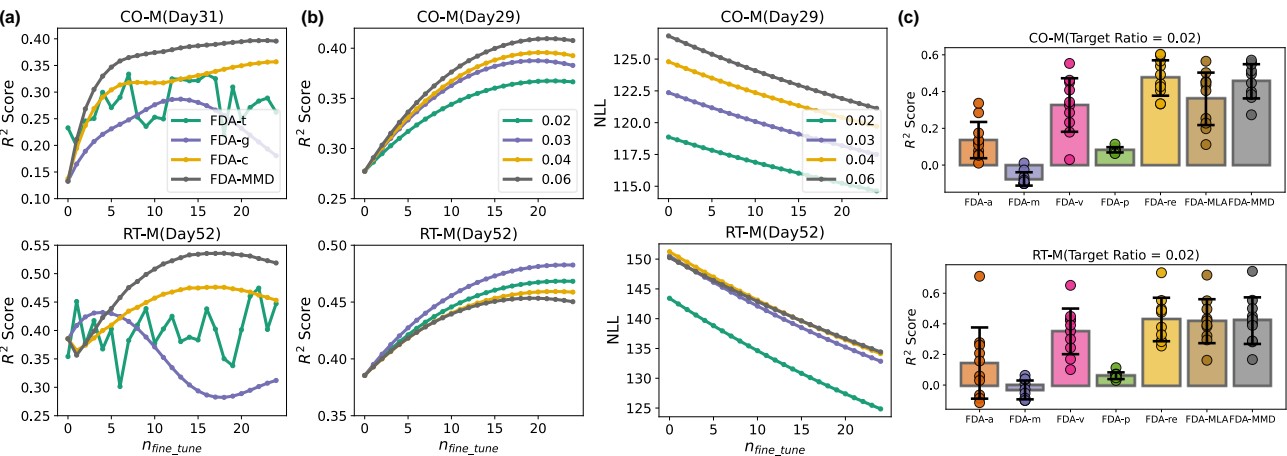

*Figure 5.* (a) $R^2$ for FDA-t, FDA-g, FDA-c, and FDA-MMD on CO-M (Day31) and RT-M (Day52) with $r = 0.02$. FDA-t extracted and aligned features solely using the transformer-based $f_\alpha$ without flows. FDA-g aligned $\mathbf{z}(1)$ using Cycle-GAN, and FDA-c aligned the distribution of $\mathbf{c}$ instead of $\mathbf{z}(1)$. (b) $R^2$ (Left) and the corresponding negative log likelihood (NLL) (Right) on CO-M (Day29) and RT-M (Day52) by FDA-MLA with various target ratios $r$. (c) Comparison of average $R^2$ scores, achieved by FDA-a, FDA-m, FDA-v, FDA-p, FDA-re, FDA-MLA, and FDA-MMD. Dots represent $R^2$ values for individual session($r = 0.02$). Bar charts denote average $R^2$ across sessions. We employed FDA-v and FDA-p as variants utilizing VP and GVP flow paths, respectively. FDA-a and FDA-m employ transformers with temporal correlation attention and MLPs as conditional extractors, respectively. FDA-re includes an additional reconstruction term to regularize the learned neural representations.

The three exceptions have MLEs below 1e-3, which can be considered approximately stable. Therefore, we conclude that the latent dynamics of our FDA method remain stable. Although ERDiff also employs a flow-based framework, its reliance on prior statistical assumptions and path implementation via transformers may compromise the stability of latent dynamics. FDA resolve this issue through learning without specific statistical assumptions via conditional flow matching. Our path implementation with simple residual MLPs also facilitates stability regularization, as demonstrated in Eq. (21). We further analyze the benefits of our representation learning based on stable latent dynamics via the zero-shot cross-session performance. As shown in Table S8, FDA achieved better performance than the best two baselines without alignment.

*Table 2.* Comparison of average $R^2$ scores (%) across sessions for FDA-al, FDA-sc, and FDA-MMD on the CO-M and RT-M datasets ($r = 0, 0.02$).

| Data | $r$ | FDA-al | FDA-sc | FDA-MMD |
|---|---|---|---|---|
| CO-M | 0 | $-9.34_{\pm 9.57}$ | $-18.20_{\pm 17.53}$ | $\mathbf{16.23_{\pm 9.43}}$ |
| | 0.02 | $14.51_{\pm 16.37}$ | $13.35_{\pm 19.01}$ | $\mathbf{45.59_{\pm 5.15}}$ |
| RT-M | 0 | $1.23_{\pm 4.76}$ | $1.78_{\pm 3.89}$ | $\mathbf{38.15_{\pm 8.21}}$ |
| | 0.02 | $16.99_{\pm 11.75}$ | $20.46_{\pm 11.77}$ | $\mathbf{42.08_{\pm 6.31}}$ |

Additionally, we performed a more comprehensive study to further validate the benefits of stable latent dynamics. Since stability is maintained by activation functions and scale coefficients, we ablated these two components (FDA-al and FDA-sc, respectively) to violate the assumptions of stability. As shown in Fig. S4, the distribution of MLEs for FDA-al and FDA-sc reveals that both variants frequently exhibit positive values, indicating instability resulting from the ab-

lation of respective components. The corresponding results for zero-shot and few-trial performance using MMD-based alignment on the CO-M and RT-M datasets are summarized in Table 2. Consistent with their reduced stability, FDA-al and FDA-sc demonstrated substantially degraded transfer performance. This result suggests that instability is the factor contributing to their poor transfer ability.

**Computational Efficiency and Hyper-parameter Analysis** We evaluated the computational efficiency of FDA compared to baselines under identical hardware configurations (NVIDIA GeForce RTX 3080 Ti, 12GB). The comparison was based on the number of parameters and training time per epoch or in total, covering both pre-training and fine-tuning phases. As shown in Table S10 and Table S11, FDA required less training time compared to ERDiff and NoMAD, owing to its efficient training objectives based on short-term context windows. Further analysis of FDA's inference time is presented in Table S12. The average inference time per window is approximately 4 ms, demonstrating its suitability for real-time applications. Moreover, the sensitivity analysis of main hyper-parameters in FDA and the level of diversity in target trials is provided in App. C.3.

**Evaluation on Simulated Neural Data** We conducted further experiments to evaluate the recovery of ground-truth latent variables in synthetic data. Following method in (Kapoor et al., 2024), we used the Lorenz attractor as the latent dynamics. We simulated firing rates as an affine transformation of the 3D latent variables into a 96-dimensional space, then sampled spike trains from a Poisson distribution. As shown in Table 3, our FDA successfully recovered the

latent dynamics from synthetic spiking data. The visualizations of our decoded 3D trajectories presented in Fig. S7 confirmed that FDA effectively captured the neural dynamics.

*Table 3.* Average $R^2$ scores(%) for recovered latent variables from synthetic spiking data at varying mean firing rates.

| Mean Firing Rates | 0.05 | 0.1 | 0.3 |
|---|---|---|---|
| $R^2$ | $95.43_{\pm0.87}$ | $95.68_{\pm1.07}$ | $95.24_{\pm1.03}$ |

### 4.3. Ablation Study

**Ablation Study on Different Alignment Strategies** To evaluate the effectiveness of our alignment, we compared FDA with several variants. FDA-t extracted and aligned features solely using the transformer-based $f_\alpha$ without flows. FDA-g aligned $\mathbf{z}(1)$ using Cycle-GAN, and FDA-c aligned the distribution of $\mathbf{c}$ instead of $\mathbf{z}(1)$. As shown in Table 4, full FDA outperformed FDA-t, whose performance fell below that of baselines, indicating the advantage of flow-based frameworks. The degraded performance of FDA-g stemmed from training instability, similar to Cycle-GAN, as illustrated in Fig. 5(a). FDA-c performed slightly worse than FDA-MMD, demonstrating the equivalence of aligning $\mathbf{c}$ and $\mathbf{z}(1)$ using MMD.

As presented in Fig. 5(a) and App. C.2.1, $R^2$ curves of FDA-MMD and FDA-c demonstrated more efficient and robust adaptation in few-trial scenarios, smoothly reaching their peak within $\sim 20$ epochs during fine-tuning. For FDA-MLA, as shown in Fig. 5(b), both the negative log-likelihood (NLL) and target $R^2$ exhibited smooth convergence within $\sim 20$ epochs across different $r$, demonstrating rapid and efficient alignment with few trials. Despite the observed decline in $R^2$ at higher $r$ values in Fig. 5(b), a general trend of increasing $R^2$ with larger fine-tuning sample sizes was observed across most sessions, as shown in Table S14 and Table S15. Furthermore, the superior performance on Day 52 (RT-M) compared to Days 29 and 31 (CO-M) can be attributed to greater similarity between the source and target sessions. This is supported by the maximum mutual information (MI) between spiking recordings from individual channels, which is higher for RT-M (2e-3) than for CO-M (6e-4).

*Table 4.* Comparison of average $R^2$ scores (in %) over sessions on CO-M and RT-M datasets. FDA-t extracted and aligned features solely using transformer-based $f_\alpha$ without flows. FDA-g aligned $\mathbf{z}(1)$ using Cycle-GAN, and FDA-c aligned the distribution of $\mathbf{c}$.

| Data | Target Ratio | FDA-t | FDA-g | FDA-c | **FDA-MMD** |
|---|---|---|---|---|---|
| CO-M | 0.02 | $-82.67_{\pm90.02}$ | $35.57_{\pm6.46}$ | $42.88_{\pm4.73}$ | $\mathbf{45.59}_{\pm5.16}$ |
| | 0.03 | $-85.45_{\pm90.20}$ | $35.23_{\pm7.45}$ | $44.69_{\pm3.72}$ | $\mathbf{48.40}_{\pm4.59}$ |
| | 0.04 | $-86.09_{\pm89.15}$ | $35.25_{\pm7.66}$ | $46.36_{\pm4.15}$ | $\mathbf{50.71}_{\pm4.68}$ |
| | 0.06 | $-55.96_{\pm85.19}$ | $34.35_{\pm8.19}$ | $47.27_{\pm4.53}$ | $\mathbf{51.10}_{\pm4.76}$ |
| RT-M | 0.02 | $0.15_{\pm23.78}$ | $40.56_{\pm7.31}$ | $\mathbf{42.28}_{\pm6.29}$ | $42.08_{\pm6.31}$ |
| | 0.03 | $1.44_{\pm24.47}$ | $40.35_{\pm7.49}$ | $43.77_{\pm6.05}$ | $\mathbf{44.36}_{\pm5.83}$ |
| | 0.04 | $3.41_{\pm18.17}$ | $40.04_{\pm7.62}$ | $44.08_{\pm6.06}$ | $\mathbf{45.35}_{\pm6.15}$ |
| | 0.06 | $3.65_{\pm16.10}$ | $39.76_{\pm7.42}$ | $46.31_{\pm4.92}$ | $\mathbf{47.23}_{\pm5.96}$ |

**Ablation Study of Main Components** Further ablation studies were conducted to validate the effectiveness of our

flow-based learning framework. The corresponding results are provided in Fig. 5(c) and Fig. S9. For the design of flow paths, we utilized FDA-v and FDA-p as variants with VP and GVP paths (Ma et al., 2024), similar to those in ERDiff. We found that the performance degradation with nonlinear flow paths may result from a mismatch in latent dynamics between the ground truth and those generated by residual MLPs, as well as the use of one-step Euler sampling. For conditional feature extractors, we used FDA-a and FDA-m as variants, incorporating transformers based on temporal correlation attention and MLPs. We observed that these extractors may not provide effective conditional features, resulting in $R^2$ degradation.

Note that baselines like NoMAD and ERDiff both incorporate a reconstruction loss term for supervised neural representation. To eliminate potential information leakage from supervised training based solely on behavioral labels, we introduced additional reconstruction term to regularize our flow-based representation (FDA-re), achieving results comparable to FDA-MMD. This further validates the advantages of our flow-based learning framework and its efficient adaptation in few-trial scenarios.

## 5. Conclusions and Limitations

In this paper, we establish a novel neural alignment framework of FDA that first leverages flow matching to achieve efficient and robust adaptation in few-trial scenarios. Our FDA approach learns flexible neural representations with stable latent dynamics, thereby facilitating source-free domain alignment through likelihood maximization. The latent dynamics of our FDA method were theoretically verified to be stable, allowing for the robust neural adaptation. Extensive experiments on motor cortex datasets demonstrate that FDA significantly enhances decoding performance for few trials. Our FDA method potentially improves the application of chronic real-world BCI deployments.

**Discussions and Limitations** This work has several limitations that warrant further investigation. First, the target sessions typically originate from the same subject, with substantial overlap in the recorded neuronal populations in this study. The effectiveness of FDA in scenarios such as cross-task or cross-subject alignment requires additional validation. Scenarios where the observed neuronal subsets across sessions may not fully overlap are also worth further investigations. Second, future studies using clinical data from human subjects (Fan et al., 2023; Karpowicz et al., 2024) could further enhance the clinical and long-term applications of BCIs. In addition to the comparison with NDT-2 (Ye et al., 2024) presented in Table S9, the effectiveness of FDA in self-supervised learning on large-scale neural data warrants further investigation. Finally, our findings indicate that FDA may be applicable to a wider range of neural recording modalities, including Neuropixels.

## Impact Statement

This paper presents Flow-based Distribution Alignment (FDA), a novel framework that leverages flow matching to enable efficient and robust adaptation with few trials. Our FDA approach learns flexible neural representations with stable latent dynamics, offering valuable insights for real-world brain-computer interface (BCI) applications. As BCI technology advances, it holds the potential to enhance quality of life and foster inclusivity. To ensure broad and equitable benefits, its development must be guided by ethical principles such as data privacy. Our paper focuses mainly on scientific research and has no obvious negative social impact.

## Acknowledgments

This work was supported by the STI 2030 Major Projects (2021ZD0200403), the Natural Science Foundation of China (62276228), and the Zhejiang Provincial Natural Science Foundation (LR24F020002).

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

# Appendix of Flow Matching for Few-Trial Neural Adaptation with Stable Latent Dynamics

## A. Method

### A.1. Detailed Architectures

We present the detailed architecture of our main modules as follows. The input neural signals have the shape of (Batch size=256, Window size=$w$, Number of channels=$m$). The latent dimensions of conditional features $\mathbf{c}$ are denoted as $k_c$, the dimension of latent states in the continuous normalizing flow as $k_z$. The dropout value is represented as $o_d$. The architectures of $f_\alpha$, and $v_\theta$ can be seen in Table S1.

*Table S1.* Detailed Architectures of FDA

| $f_\alpha$ | [MSA($k_c, n_{head}$), FFN($k_c \times n_{head}, k_c$)]$\times 2$ |
|---|---|
| $v_\theta$ | MLP($k_z + k_c, k_z, v_d$)$\times 5$ |

Here, we use the term MLP to refer to Multilayer perceptron with residual connections, MSA to represent multi-head self-attention modules, and FFN to indicate feed-forward neural networks.

Moreover, default dimensions $k_c$, $k_z$, the drop-out rate $v_d$, the number of heads $n_{head}$, and the window length $w$ mentioned above are configured as shown in Table S2 according to different datasets.

*Table S2.* Default Value Setup on Different Datasets

|  | $k_c$ | $k_z$ | $v_d$ | $n_{head}$ | $w$ |
|---|---|---|---|---|---|
| CO-C | 64 | 64 | 0.1 | 8 | 6 |
| CO-M | 32 | 32 | 0.1 | 8 | 5 |
| RT-M | 32 | 32 | 0.1 | 8 | 5 |

### A.2. Proof of Dynamical Stability in Theorem 3.1

- First, consider the iterative relationship between two sampling steps. For example, analyzing the upper bound of $\|z_i^S(\tau_1) - z_j^S(\tau_1)\|$ is as follows:

$$\|z_i^S(\tau_1) - z_j^S(\tau_1)\| = \|z_i^S(0) + v_\theta(z_i^S(0), 0, f_\alpha(x_i^S)) - z_j^S(0) - v_\theta(z_j^S(0), 0, f_\alpha(x_i^S))\| \tag{12}$$

$$\leq \|z_i^S(0) - z_j^S(0)\| + \|v_\theta(z_i^S(0), 0, f_\alpha(x_i^S)) - v_\theta(z_j^S(0), 0, f_\alpha(x_i^S))\|. \tag{13}$$

In this study, we use an MLP layers with residuals to compose $v_\theta$ as illustrated in (Ma et al., 2024), leading to:

$$v_\theta(z_i^S(0), 0, f_\alpha(x_i^S)) \approx (2 + \gamma_i^S)z_i^S(0) + \beta_i^S, \tag{14}$$

where $\gamma_i^S$ is the scale coefficient, and we assume $0 < \|3 + \gamma_i^S\| < 1$. We only consider the influence of $f_\alpha(x_i^S)$ on $\gamma_i^S$ due to the same sampling time point:

$$\gamma_i^S = g(\mathbf{w}_\gamma f_\alpha(x_i^S) + \mathbf{b}_\gamma). \tag{15}$$

Similarly, $\beta_i^S$ is calculated in the same way:

$$\beta_i^S = g(\mathbf{w}_\beta f_\alpha(x_i^S) + \mathbf{b}_\beta). \tag{16}$$

Thus:

$$v_\theta(z_j^S(0), 0, f_\alpha(x_j^S)) \approx (2 + \gamma_j^S)z_j^S(0) + \beta_j^S. \tag{17}$$

Substituting the expansions of $v_\theta$ into the earlier equation yields:

$$\|z_i^S(\tau_1) - z_j^S(\tau_1)\| \leq \|z_i^S(0) - z_j^S(0)\| + \|(2 + \gamma_i^S)z_i^S(0) - (2 + \gamma_j^S)z_j^S(0)\| + \|\beta_i^S - \beta_j^S\| \tag{18}$$

$$\approx \|3 + \gamma_i^S\| \|z_i^S(0) - z_j^S(0)\| + \|\beta_i^S - \beta_j^S\|. \tag{19}$$

Further expanding $\|\beta_i^S - \beta_j^S\|$:

$$\|\beta_i^S - \beta_j^S\| = \|g(\mathbf{w}_\beta f_\alpha(x_i^S) + \mathbf{b}_\beta) - g(\mathbf{w}_\beta f_\alpha(x_j^S) + \mathbf{b}_\beta)\|. \tag{20}$$

Since the activation function $g$ of the MLP is typically a Lipschitz continuous function (e.g., sigmoid function), this simplifies to:

$$\|\beta_i^S - \beta_j^S\| \leq \mathbf{K}_g \|\mathbf{w}_\beta\| \|f_\alpha(x_i^S) - f_\alpha(x_j^S)\| = \mathbf{K}_g \|\mathbf{w}_\beta\| \|c_i^S - c_j^S\|, \tag{21}$$

where $\mathbf{K}_g$ is the Lipschitz constant of the function $g$. Therefore:

$$\|z_i^S(\tau_1) - z_j^S(\tau_1)\| \leq \mathbf{K}_\gamma \|z_i^S(0) - z_j^S(0)\| + \mathbf{K}_g \|\mathbf{w}_\beta\| \|c_i^S - c_j^S\|, \tag{22}$$

where $0 < \mathbf{K}_\gamma = \|3 + \gamma_i^S\| < 1$.

- Next, substituting $t_n$ into the above Eq. (22), we obtain the approximate upper bound for $\|z_i^S(\tau_n) - z_j^S(\tau_n)\|$:

$$\|z_i^S(\tau_n) - z_j^S(\tau_n)\| \leq (\mathbf{K}_\gamma)^n \|z_i^S(0) - z_j^S(0)\| + \left[\sum_{a=1}^{n-1} (\mathbf{K}_\gamma)^a\right] \mathbf{K}_g \|w_\beta\| \|c_i^S - c_j^S\|. \tag{23}$$

Let $h_z(\|z_i^S(0) - z_j^S(0)\|, n) = (\mathbf{K}_\gamma)^n \|z_i^S(0) - z_j^S(0)\|$, where $h_z : \mathbb{R}_{\geq 0} \times \mathbb{Z}_{\geq 0} \to \mathbb{R}_{\geq 0}$ is a decreasing function with respect to $n$.

Let $h_c(\|c_i^S - c_j^S\|) = \left[\sum_{a=1}^{n-1} (\mathbf{K}_\gamma)^a\right] \mathbf{K}_g \|w_\beta\| \|c_i^S - c_j^S\|$, where $h_c : \mathbb{R}_{\geq 0} \to \mathbb{R}_{\geq 0}$, and $h_c(\|c_i^S - c_j^S\|) \to \infty$ as $\|c_i^S - c_j^S\| \to \infty$.

- In summary, the latent space extracted by our method exhibits the dynamical stability defined in (Angeli, 2002).

### A.3. General computation of likelihood in Section 3.2.2

More generally, alternative sampling methods can employ the unbiased Hutchinson-trace estimator (Hutchinson, 1989) to estimate the divergence in Eq. (2). The detailed computation is presented below.

Using the instantaneous change of variables formula (Chen et al., 2018), the log-likelihood $\log p_1(\mathbf{z}^T(1)|f_\alpha(\mathbf{x}^T))$ can be expressed as:

$$\log p_1(\mathbf{z}^T(1)|f_\alpha(\mathbf{x}^T)) = \log p_0(\mathbf{z}^T(0)|f_\alpha(\mathbf{x}^T)) - \int_0^1 \nabla \cdot v_\theta(\mathbf{z}^T(\tau), f_\alpha(\mathbf{x}^T), \tau) \, d\tau, \tag{24}$$

where the latent variable $\mathbf{z}^T(\tau)$ can be calculated using any sampling method based on Eq. (1). Furthermore, we estimate $\nabla \cdot v_\theta(\mathbf{z}^T(\tau), f_\alpha(\mathbf{x}^T), \tau)$ via the unbiased Hutchinson-trace estimator.

Specifically, $\nabla \cdot v_\theta(\mathbf{z}^T(\tau), f_\alpha(\mathbf{x}^T), \tau)$ is estimated as:

$$\nabla \cdot v_\theta(\mathbf{z}^T(\tau), f_\alpha(\mathbf{x}^T), \tau) = \mathbb{E}_{p(\epsilon)}[\epsilon^\top \nabla v_\theta(\mathbf{z}^T(\tau), f_\alpha(\mathbf{x}^T), \tau)\epsilon], \tag{25}$$

where $\nabla v_\theta(\mathbf{z}^T(\tau), f_\alpha(\mathbf{x}^T), \tau)$ can be computed via reverse-mode automatic differentiation. The random variable $\epsilon$ satisfies $\mathbb{E}_{p(\epsilon)}[\epsilon] = 0$ and $\text{Cov}_{p(\epsilon)}[\epsilon] = I$.

### A.4. Related Work on Normalizing Flows

Normalizing flows have been widely applied in distribution sampling due to their precise and explicit likelihood modeling. Traditional normalizing flows (Chen et al., 2019; Dinh et al., 2022) typically rely on invertible transformations, but these can constrain the representational capacity of the networks. Recent research has sought to alleviate this limitation by utilizing continuous normalizing flows (Yang et al., 2019) based on ODEs. For example, flow matching (Liu et al., 2023a; Lipman et al., 2023; Ma et al., 2024) extends diffusion models, an advanced generative model, allowing for more flexible diffusion paths. Conditional flow matching (Liu et al., 2023b; Atanackovic et al.) further incorporates conditional features to model conditional distributions.

# B. Experimental Details

## B.1. Dataset Description

**CO-C&CO-M**(Ma et al., 2023). Monkeys C and M conducted a center-out (CO) reaching task while holding an upright handle. Monkey C utilized its right hand, whereas Monkey M used its left. Each trial commenced with the monkey positioning its hand at the center of the workspace. After a random delay, one of eight evenly spaced outer targets arranged in a circle was displayed. The monkey then maintained its position through a variable pause until hearing an auditory go cue. To earn a liquid reward, the monkey needed to reach the outer target within 1.0 second and sustain its hold for 0.5 seconds.

**RT-M**(Ma et al., 2023). Monkey M also participated in a random-target (RT) task, where it reached for sequences of three targets shown in random locations on the screen. This task utilized the same apparatus as the CO reaching task. Each trial started with the monkey placing its hand at the center of the workspace, followed by the sequential presentation of three targets. The monkey had 2.0 seconds to move the cursor to each target after seeing it. Due to the random positioning of the targets, the cursor trajectory varied with each trial.

**Preprocess Process**. For all datasets, we extracted trials from the 'go cue time' to the 'trial end.' Next, we processed the neural signals by digitizing, applying a bandpass filter (250-5000 Hz), and detecting spikes using thresholds based on root-mean square activity. The data was then timestamped and smoothed with a Gaussian kernel to compute firing rates over 50 ms bins.

## B.2. Training Details

The main configurations for model training included the learning rate, weight decay parameters of the Adam optimizer, batch sizes, number of iterative epochs during pre-training and fine-tuning phases. Details of these hyperparameters are provided in Table S3 and Table S4, respectively.

*Table S3.* Detailed Pre-training Setup

|      | Learning Rate | Weight Decay | Epochs | Batch Size |
|------|---------------|--------------|--------|------------|
| CO-C | 2e-3          | 1e-5         | 3500   | 256        |
| CO-M | 2e-3          | 1e-5         | 3500   | 256        |
| RT-M | 2e-3          | 1e-5         | 3500   | 256        |

*Table S4.* Detailed Fine-tuning Setup

|      | Learning Rate | Weight Decay | Epochs | Batch Size |
|------|---------------|--------------|--------|------------|
| CO-C | 1e-4          | 1e-5         | 25     | 256        |
| CO-M | 1e-4          | 1e-5         | 25     | 256        |
| RT-M | 1e-4          | 1e-5         | 25     | 256        |

## B.3. Baseline Implementation

**CEBRA**(Schneider et al., 2023). CEBRA is a sophisticated machine-learning approach aimed at analyzing and compressing time series data, particularly in the context of behavioral and neural studies. It excels at revealing hidden structures in data variability and has been effectively applied to decode neural activity in the mouse brain's visual cortex, allowing for the reconstruction of what the subject has seen. The code can be accessed at https://github.com/AdaptiveMotorControlLab/cebra.

**ERDiff**(Wang et al., 2023b). ERDiff introduces a method that utilizes diffusion models to extract latent dynamic structures from the source domain and subsequently recover them in the target domain using maximum likelihood alignment. Empirical evaluations on both synthetic and neural recording datasets indicate that this approach surpasses others in effectively preserving latent dynamic structures over time and across individuals. The latest code can be accessed at https://github.com/yulewang97/ERDiff.

**NoMAD**(Karpowicz et al., 2022). NoMAD utilizes the latent manifold structure present in neural population activity to create a reliable connection between brain activity and motor behavior. It shows the capability to achieve accurate and

highly stable behavioral decoding over long durations, thus eliminating the necessity for supervised recalibration. In this study, we implemented NoMAD using the LFADS code found at https://github.com/arsedler9/lfads-torch/tree/main, which may lead to some differences from the original implementation.

**Cycle-GAN**(Ma et al., 2023). Cycle-GAN aligned the distributions of full-dimensional neural recordings, stabilizing the original decoding model without the need for recalibration. Evaluations of Cycle-GAN alongside a related approach (ADAN) on multiple monkey and task datasets reveal that Cycle-GAN outperforms in maintaining BCI accuracy robustly over time without additional training. Since this study employs the same datasets, we directly implement the publicly available code from https://github.com/limblab/adversarial_BCI.

### B.4. Validation Details

Specifically, during the validation after fine-tuning phases, we employed neural signals $\mathbf{x}^T$ from the target domain, which were not leveraged during the fine-tuning phase, to evaluate the efficacy of our alignment approach.

This evaluation is based on the decoding performance based on $R^2$ scores. We first sample $\mathbf{z}^T(1)$ using the one-step Euler based on $\mathbf{z}^T(0)$: $\mathbf{z}^T(1) = v_\theta(\mathbf{z}^S(0), 0, f_\alpha(\mathbf{x}^S))$. The predicted target label $\tilde{\mathbf{y}}^T$ are computed as below: $\tilde{\mathbf{y}}^T = \eta^* \mathbf{z}^T(1)$. $R^2$ scores are further obtained between $\tilde{\mathbf{y}}^T$ and actual $\mathbf{y}^T$.

### B.5. Lyapunov Exponents

The stability described above can be quantified using the Lyapunov function (Angeli, 2002), which can also be estimated through the maximum Lyapunov exponent (MLE). The maximum Lyapunov exponent $\lambda$ can be defined based on the latent state $\mathbf{z}(t)$ as follows: $\lambda = \lim_{t \to \infty} \lim_{|\delta\mathbf{z}(0)| \to 0} \frac{1}{t} \ln \frac{|\delta\mathbf{z}(t)|}{|\delta\mathbf{z}(0)|}$. A non-positive MLE often indicates the stability of dynamical systems, achieving stable dynamical latent features (Wolf et al., 1985). Here, we estimated the MLE $\lambda$ of $z_i$ based on the method in (Wolf et al., 1985) to evaluate the stability of dynamical latent features extracted from $\mathcal{D}_S$ after the pre-training phase. The detailed calculation of $\lambda$ is available below.

The stability defined in (Angeli, 2002) can be determined using a Lyapunov function $V(z)$: given an equilibrium point $z^*$ of the system,
$V(z^*) = 0$,
$\dot{V}(z^*) = 0$,
$V(z) > 0$ for all $z \neq z^*$,
$\dot{V}(z) < 0$ for all $z \neq z^*$.

It is known that $V(z) = \frac{1}{2}z^T z$ is one of the functions that meet the conditions. However, directly calculating complex $V(z)$ can be difficult. Therefore, we used the method based on (Wolf et al., 1985) to estimate the stability of $z(t)$ as follows:

**Step 1:**
Select $N$ sample points, denoted one as $z_1(t_0)$, find $j$ such that $j = \arg\min_k \|z_1(t_0) - z_k(t_0)\|$, and let $L_0(t_0) = \|z_1(t_0) - z_j(t_0)\|$.

**Step 2:**
Find $t_i$, for a given constant $\epsilon$, such that $t_0 \leq t < t_i$, $L_0(t) \leq \epsilon$; $L_0(t_i) > \epsilon$. Let $L'_0 = L_0(t_i)$. Continue with $z_1(t_i)$ as the next sample point following Step 1.

**Step 3:**
The maximum Lyapunov exponent(MLE) $\lambda$ is approximately as follows:

$$\lambda \approx \frac{1}{N\Delta t} \sum_{s=1}^{M} \log_2 \left( \frac{L'_0}{L_0(t_0)} \right),$$

where $\Delta t$ is the time step interval and $M$ is the number of steps in a single orbit.

# C. Additional Results

## C.1. Comparative Study

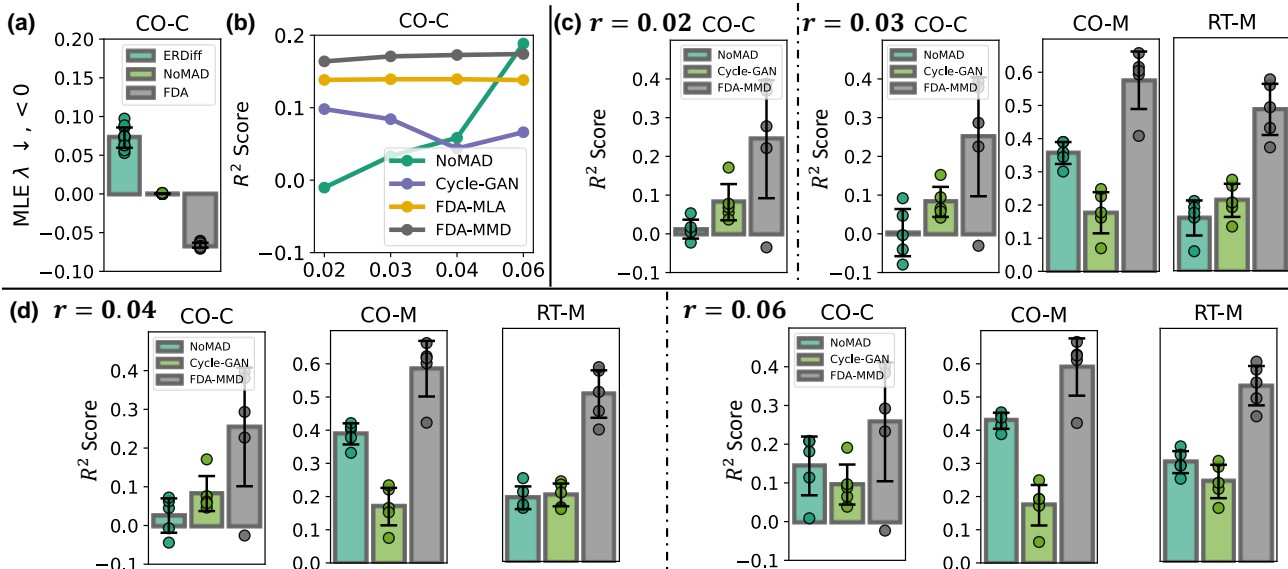

*Figure S1.* (a) The maximum Lyapunov exponent (MLE) $\lambda$ achieved by ERDiff, NoMAD, and FDA is displayed for the CO-C dataset. Dots in different colors represent the average MLE from individual sessions. (b) Average $R^2$ scores for NoMAD, Cycle-GAN, FDA-MLA, and FDA-MMD are presented under varying values of $r$ on CO-C. (c) and (d): $R^2$ scores for cross-session decoding ($r = 0.02, 0.03$ (c) and $r = 0.04, 0.06$ (d)) when $\mathcal{D}_S$ contains two sessions, obtained from NoMAD, Cycle-GAN, and FDA-MMD, are shown. Dots in different colors represent the average $R^2$ scores for different $\mathcal{D}_S$.

### C.1.1. LATENT DYNAMICS STABILITY

To validate the dynamical stability of FDA, we measured the maximum Lyapunov exponent (MLE) $\lambda$ of $\mathbf{z}^S(\tau)$ after pre-training on $\mathcal{D}_S$. The value of $\lambda$ was computed as described in (Wolf et al., 1985), and the results of CO-C is shown in Fig. S1(a).

Moreover, we also visualized all maximum Lyapunov exponents (MLE) achieved by ERDiff, NoMAD, and FDA across target sessions. As shown in Fig. S2(a), FDA consistently achieved non-positive MLEs in most cases, aligning with the average MLE results. Compared to ERDiff and NoMAD, the latent dynamics of our learning based on FDA are more stable.

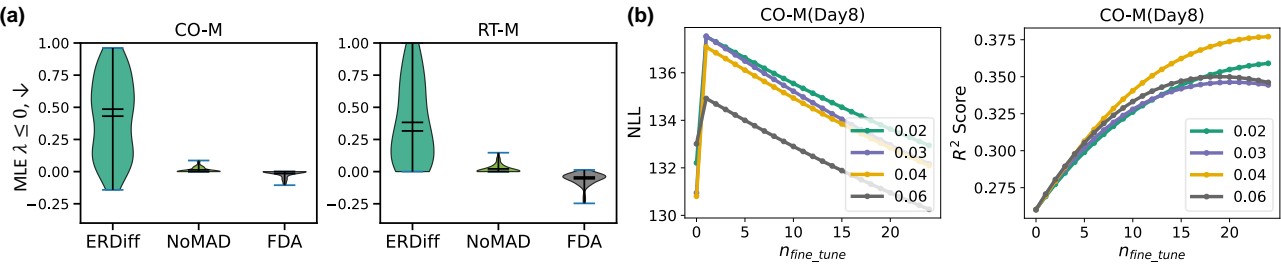

*Figure S2.* (a) Violin plot of all maximum Lyapunov exponents (MLE) $\lambda$ achieved by ERDiff, NoMAD, and FDA on the CO-M and RT-M datasets, with a non-positive $\lambda$ typically indicating dynamical stability. (b) Negative log likelihood (NLL) (Left) and the corresponding $R^2$ (Right) curves on CO-M (Day8) by FDA-MLA with various target ratios $r$.

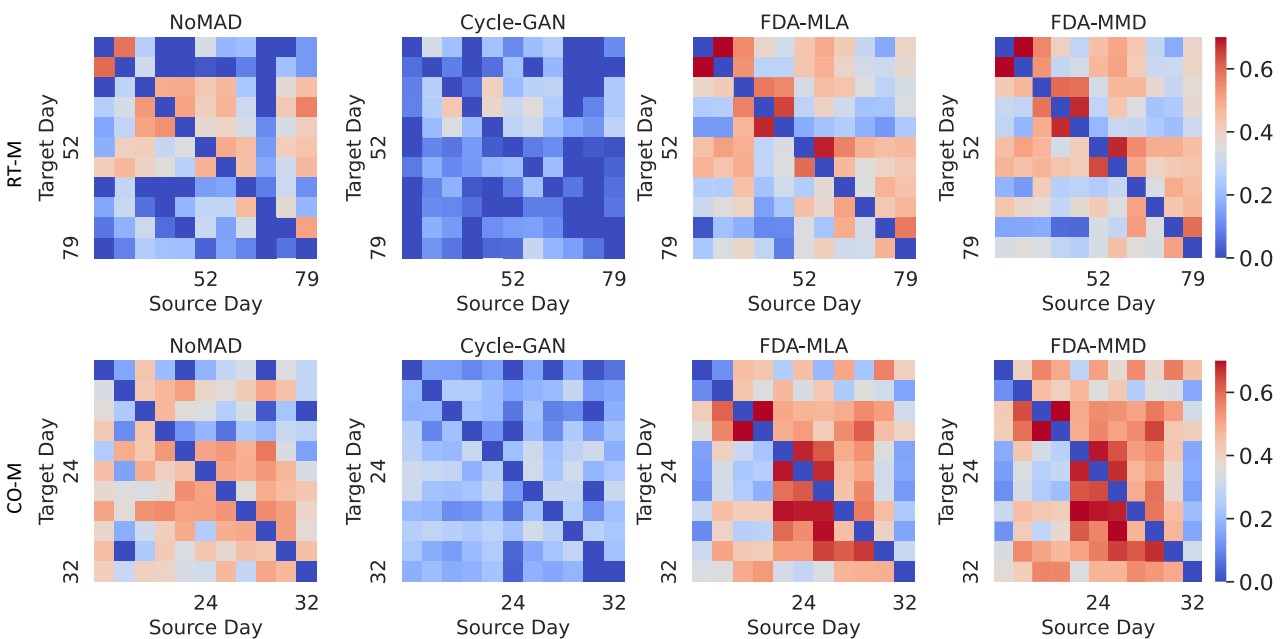

*Figure S3.* Overall performance of average $R^2$ scores ($r = 0.02$) for NoMAD, Cycle-GAN, FDA-MLA, and FDA-MMD are demonstrated on RT-M, and CO-M datasets. Blocks with various colors represent the corresponding values of $R^2$. For convenience, the diagonal values of the in-session performance were set to 0 by default.

Additionally, we conducted a more thorough ablation study to validate the assumption of stability. Since stability is governed by activation functions and scale coefficients, we ablated these two components individually (FDA-al and FDA-sc, respectively) to violate the assumption. As shown in Fig. S4, the distribution of maximum Lyapunov exponents (MLE) for FDA-al and FDA-sc reveals frequent occurrences of positive MLEs, indicating instability.

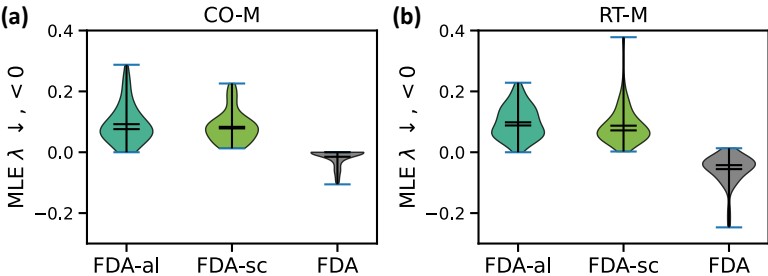

*Figure S4.* Violin plot of all maximum Lyapunov exponents (MLE) $\lambda$ achieved by FDA-al(FDA without activation functions), FDA-sc(FDA without scale coefficients) and FDA on the (a) CO-M and (b) RT-M datasets, with a non-positive $\lambda$ typically indicating dynamical stability.

C.1.2. CROSS-SESSION PERFORMANCE

We verified the cross-session performance of FDA with limited target trials. First, we conducted experiments with $\mathcal{D}_S$ containing only one session. The full average $R^2$ scores on the CO-C dataset, using Day0 as the source session and a target ratio $r$ of 0.02, are presented in Table S5.

In addition, as illustrated in Fig. S1(b), FDA achieved significantly higher average $R^2$ scores across different values of $r$. The overall performance of average $R^2$ on RT-M, and CO-M datasets is presented in Fig. S3.

More comparisons on all datasets when $\mathcal{D}_S$ included two sessions when $r$ equals 0.02, 0.03, 0.04, and 0.06 are shown in Fig. S1(c) and (d).

*Table S5.* Comparison of $R^2$ values (in %) of baselines and FDA on the CO-C dataset($r = 0.02$).

| Data | Session | LSTM | CEBRA | ERDiff | NoMAD | Cycle-GAN | **FDA-MLA** | **FDA-MMD** |
|------|---------|------|-------|--------|-------|-----------|-------------|-------------|
| CO-C | Day 0 | $86.65_{\pm1.18}$ | $87.86_{\pm0.98}$ | $\mathbf{88.69}_{\pm0.74}$ | $87.99_{\pm3.45}$ | $84.54_{\pm1.32}$ | $81.63_{\pm2.88}$ | $81.63_{\pm2.88}$ |
| | Day 1 | $5.04_{\pm27.90}$ | $18.87_{\pm6.82}$ | $-0.34_{\pm0.03}$ | $27.76_{\pm5.21}$ | $8.05_{\pm9.86}$ | $49.13_{\pm5.03}$ | $\mathbf{50.84}_{\pm5.32}$ |
| | Day 2 | $9.25_{\pm32.85}$ | $\mathbf{44.73}_{\pm14.03}$ | $-0.49_{\pm0.04}$ | $34.81_{\pm3.40}$ | $15.35_{\pm11.34}$ | $36.25_{\pm5.60}$ | $34.28_{\pm5.35}$ |
| | Day 3 | $-128.25_{\pm65.07}$ | $\mathbf{24.47}_{\pm7.60}$ | $-0.39_{\pm0.05}$ | $17.59_{\pm6.71}$ | $5.40_{\pm7.21}$ | $7.54_{\pm4.52}$ | $8.49_{\pm3.85}$ |
| | Day 9 | $-24.15_{\pm33.53}$ | $7.79_{\pm23.55}$ | $-0.25_{\pm0.02}$ | $31.07_{\pm3.93}$ | $18.37_{\pm7.71}$ | $\mathbf{38.02}_{\pm7.84}$ | $33.22_{\pm7.69}$ |
| | Day 10 | $-70.33_{\pm65.25}$ | $14.64_{\pm3.55}$ | $-0.97_{\pm0.10}$ | $\mathbf{30.05}_{\pm6.07}$ | $20.30_{\pm8.84}$ | $1.21_{\pm2.61}$ | $0.76_{\pm1.26}$ |
| | Day 14 | $-65.46_{\pm24.55}$ | $-12.97_{\pm41.24}$ | $-0.62_{\pm0.06}$ | $\mathbf{29.10}_{\pm1.57}$ | $2.67_{\pm14.34}$ | $22.99_{\pm7.08}$ | $16.40_{\pm8.49}$ |
| | Day 15 | $-32.08_{\pm24.64}$ | $-12.95_{\pm27.23}$ | $-0.53_{\pm0.02}$ | $\mathbf{21.72}_{\pm5.40}$ | $19.55_{\pm16.31}$ | $9.80_{\pm15.59}$ | $15.35_{\pm12.25}$ |
| | Day 16 | $-123.74_{\pm63.89}$ | $-9.18_{\pm30.96}$ | $-0.44_{\pm0.03}$ | $9.32_{\pm4.15}$ | $6.70_{\pm11.45}$ | $5.09_{\pm8.98}$ | $\mathbf{11.04}_{\pm6.03}$ |
| | Day 36 | $-70.67_{\pm99.37}$ | $-30.76_{\pm30.03}$ | $-0.33_{\pm0.06}$ | $-5.76_{\pm3.74}$ | $-9.40_{\pm16.54}$ | $-4.81_{\pm6.74}$ | $\mathbf{0.99}_{\pm2.62}$ |
| | Day 37 | $-29.54_{\pm59.36}$ | $-21.54_{\pm29.56}$ | $-0.40_{\pm0.06}$ | $8.40_{\pm1.36}$ | $8.76_{\pm6.63}$ | $3.08_{\pm9.33}$ | $\mathbf{15.95}_{\pm5.73}$ |
| | Day 38 | $-112.02_{\pm132.39}$ | $-7.36_{\pm16.59}$ | $-0.48_{\pm0.05}$ | $6.88_{\pm5.40}$ | $12.17_{\pm7.03}$ | $-2.77_{\pm8.46}$ | $\mathbf{12.95}_{\pm0.92}$ |

To explore the differences in results between Monkey C and Monkey M, we analyzed the cross-session performance of FDA-MMD with greater target ratios $r$. As shown in Table S6, although FDA-MMD initially performed worse on CO-C, its performance improved significantly and became comparable to RT-M when $r$ exceeded 0.3 (approximately 60 trials). Additionally, we observed larger deviations per session on CO-C. This suggests that the difference arises from instability caused by outliers specifically from the dataset of Monkey C, which notably impacted performance when $r$ was small.

*Table S6.* Comparison of average $R^2$ values (%) across sessions for FDA-MMD on the CO-C, CO-M, and RT-M datasets ($r = 0.02$). The average standard deviations over five runs per session are also reported.

| $r$ | **0.02** | **0.03** | **0.04** | **0.06** | **0.1** | **0.2** | **0.3** | **0.4** | **0.5** | **0.6** |
|-----|----------|----------|----------|----------|---------|---------|---------|---------|---------|---------|
| CO-C | $16.40_{\pm5.40}$ | $17.08_{\pm7.53}$ | $17.27_{\pm8.58}$ | $17.41_{\pm7.66}$ | $28.18_{\pm5.36}$ | $42.61_{\pm5.23}$ | $50.12_{\pm6.90}$ | $54.87_{\pm5.05}$ | $55.05_{\pm5.71}$ | $56.00_{\pm4.88}$ |
| CO-M | $45.59_{\pm5.15}$ | $48.40_{\pm4.59}$ | $50.71_{\pm4.68}$ | $51.10_{\pm4.76}$ | $57.90_{\pm2.68}$ | $62.20_{\pm2.41}$ | $65.16_{\pm2.53}$ | $66.38_{\pm2.44}$ | $66.78_{\pm2.48}$ | $67.32_{\pm3.32}$ |
| RT-M | $42.08_{\pm6.31}$ | $44.36_{\pm5.83}$ | $45.35_{\pm6.15}$ | $47.23_{\pm5.96}$ | $52.15_{\pm4.16}$ | $53.66_{\pm3.35}$ | $55.28_{\pm2.89}$ | $56.45_{\pm2.89}$ | $56.53_{\pm2.55}$ | $57.93_{\pm2.39}$ |

Additionally, we observed that the worst $R^2$ score occurred on different days for each method. This variability may stem from the different criteria used for optimal alignment. For instance, FDA-MLA exhibited an abnormal increase in NLL during the initial fine-tuning epochs on Day 8 (CO-M), as shown in Fig. S2(b). In contrast, other methods, such as NoMAD based on KL divergences and LSTM without alignment, did not show this phenomenon on the same day, leading to the worst performance of FDA-MLA while others did not experience such an issue.

C.1.3. CROSS-SESSION PERFORMANCE UNDER DIFFERENT LATENT DIMENSIONS

To determine the appropriate latent dimensions, we conducted experiments on CEBRA under varying latent dimensions. As shown in Table S7, we selected the latent dimensions for CEBRA as 32, based on its better performance. For ERDiff and NoMAD, we set the latent dimension to 8 and 16 respectively, following the default settings mentioned in the original paper due to its application to similar datasets.

*Table S7.* Average $R^2$ scores across sessions of CEBRA on CO-M and RT-M datasets under different latent dimensions.

| Latent Dimension | 16 | 32 | 48 |
|---|---|---|---|
| CO-M | $-1.34_{\pm 11.69}$ | $\mathbf{1.14}_{\pm 14.47}$ | $0.85_{\pm 12.61}$ |
| RT-M | $-53.01_{\pm 14.49}$ | $\mathbf{-45.48}_{\pm 12.51}$ | $-49.21_{\pm 14.71}$ |

### C.1.4. ZERO-SHOT CROSS-SESSION PERFORMANCE

Additionally, we compared the zero-shot cross-session performance of NoMAD without alignment, Cycle-GAN without alignment, and FDA without alignment, with detailed results presented in Table S8. FDA without fine-tuning outperformed the baselines, which we attribute to the dynamical stability of its pre-trained latent spaces. Furthermore, performance in few-trial scenarios continued to improve after fine-tuning. In summary, the combination of stable latent dynamics and efficient fine-tuning contributes to FDA's better performance in few-trial scenarios.

*Table S8.* Comparison of $R^2$ values (in %) across target sessions (where the $R^2$ scores for each session are averaged over five random runs with different sample selections) of baselines and FDA without alignment on CO-M and RT-M datasets.

| Data | NoMAD w/o alignment | Cycle-GAN w/o alignment | **FDA w/o alignment** | FDA-MLA | FDA-MMD |
|---|---|---|---|---|---|
| CO-M | $-121.47_{\pm 77.80}$ | $-126.84_{\pm 23.82}$ | $16.23_{\pm 9.43}$ | $36.05_{\pm 5.84}$ | $45.59_{\pm 5.15}$ |
| RT-M | $-74.06_{\pm 49.94}$ | $-3.42_{\pm 5.55}$ | $38.15_{\pm 8.21}$ | $41.73_{\pm 4.88}$ | $42.08_{\pm 6.31}$ |

### C.1.5. ZERO-SHOT PERFORMANCE COMPARISON WITH NDT-2

We utilized 47 sessions recorded from the motor cortex of two monkeys, available via the external link (https://zenodo.org/records/3854034), as well as datasets provided by the Neural Latents Benchmark (https://neurallatents.github.io/) for pre-training NDT-2 (Ye et al., 2024). Subsequently, supervised fine-tuning was conducted using 80% of the trials from a source session of the CO-C/CO-M datasets, followed by zero-shot evaluation on the remaining target sessions.

The average $R^2$ scores are presented in Table S9. While NDT-2 achieved higher performance in intra-session decoding on the source session, its performance degraded substantially during zero-shot evaluation. This suggests that NDT-2 may be more effective when fine-tuned with supervised data from the target sessions.

*Table S9.* Average $R^2$ scores (%) of NDT-2 for intra-session and inter-session decoding on the CO-C and CO-M datasets.

| Data | Intra-session | Inter-session |
|---|---|---|
| CO-C | $89.82_{\pm 1.18}$ | $-34.70_{\pm 0.32}$ |
| CO-M | $93.75_{\pm 1.51}$ | $-52.31_{\pm 0.16}$ |

### C.1.6. PERFORMANCE WITH DIFFERENT TARGET RATIOS $r$

To further evaluate the performance of FDA under different target ratios $r$, we gradually increased $r$ from 0.02 to 0.6. The $R^2$ scores for NoMAD, Cycle-GAN, and FDA are shown in Fig. S5. In particular, Cycle-GAN and NoMAD exhibited significantly lower performance (approximately five times worse) with fewer target samples. However, as $r$ increased to around 0.3 (approximately 60 trials), their performance became comparable to that of FDA-MLA and FDA-MMD.

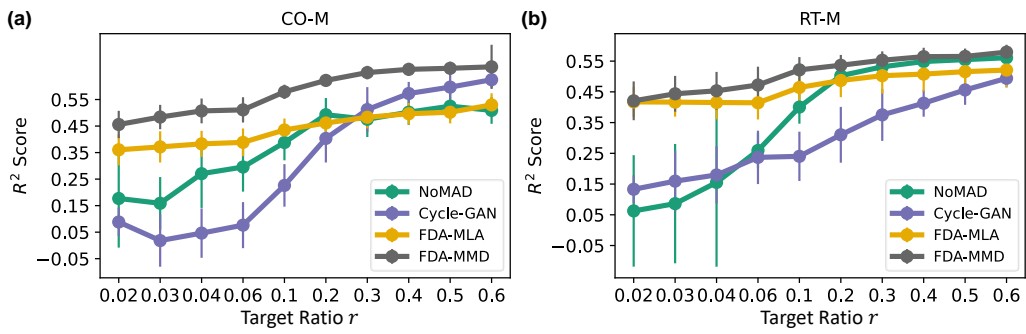

*Figure S5.* Comparison of average $R^2$ scores across target sessions (where the $R^2$ scores for each session are averaged over five random runs with different sample selections) for NoMAD, Cycle-GAN, FDA-MLA, and FDA-MMD under different target ratios $r$ on the (a) CO-M and (b) RT-M datasets.

Additionally, we examined the $R^2$ curves across target sessions for FDA-MMD and Cycle-GAN on the CO-M dataset. As shown in Fig. S6, both methods exhibited fluctuating $R^2$ curves at small target ratios. However, as the target ratio increased, the fluctuations were alleviated. With the exception of a few sessions, $R^2$ scores generally decreased across most target sessions. We attribute this trend to the reduced influence of certain outliers in scenarios with few target samples.

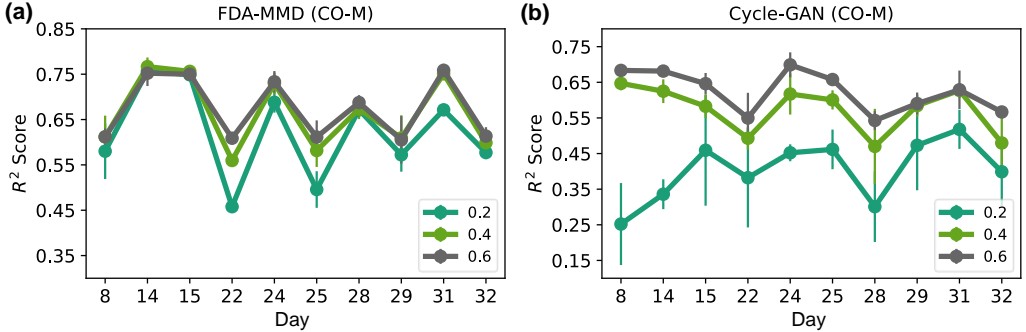

*Figure S6.* $R^2$ curves across target sessions for (a) FDA-MMD and (b) Cycle-GAN under different target ratios $r$ (0.2, 0.4, and 0.6) on the CO-M dataset.

### C.1.7. COMPUTATIONAL EFFICIENCY

We compared the computational efficiency of our methods with that of ERDiff, Cycle-GAN, and NoMAD on NVIDIA GeForce RTX 3080 Ti (12GB). The comparison was based on the number of parameters and training time per epoch, which includes pre-training and fine-tuning, on CO-C, CO-M, and RT-M. As shown in Table S10, FDA-MLA and FDA-MMD exhibited a higher number of parameters. However, they required less training time compared to ERDiff and NoMAD, which can be attributed to effective training losses and sampling methods.

*Table S10.* The computational efficiency comparison between the baselines and FDA, evaluated in terms of the number of parameters and per-epoch training time (s), including both pre-training and fine-tuning, across the CO-C, CO-M, and RT-M datasets.

| | Method | ERDiff(Wang et al., 2023b) | Cycle-GAN(Ma et al., 2023) | NoMAD(Karpowicz et al., 2022) | FDA-MLA | FDA-MMD |
|---|---|---|---|---|---|---|
| | Parameter Number (M) | 0.04 | 0.03 | 0.05 | 0.03 | 0.03 |
| Time(s) | CO-C | 0.39 | 0.05 | 1.05 | 0.14 | 0.14 |
| | CO-M | 1.14 | 0.02 | 1.03 | 0.13 | 0.14 |
| | RT-M | 0.49 | 0.02 | 1.04 | 0.10 | 0.10 |

A more detailed comparison of total training time (in seconds) during the pre-training and fine-tuning phases is provided in Table S11.

*Table S11.* Total training time (s) for pre-training and fine-tuning phases across baselines and FDA

| Phase | ERDiff | Cycle-GAN | NoMAD | FDA-MLA | FDA-MMD |
|---|---|---|---|---|---|
| Pre-training | 2449.68 | - | 135.34 | 98.95 | 98.95 |
| Fine-tuning | 1.21 | 11.90 | 76.83 | 2.32 | 2.38 |

We further analyzed the inference time of FDA-MLA and FDA-MMD on an NVIDIA GeForce GTX 1080 Ti (11GB). As shown in the table below, the average inference time for a single window is approximately 4 ms, indicating the method's suitability for real-time applications.

*Table S12.* Average inference time (ms) of FDA-MLA and FDA-MMD.

| Data | FDA-MLA | FDA-MMD |
|---|---|---|
| Avg | 3.90 | 3.97 |

### C.1.8. EXPERIMENTS ON SYNTHETIC NEURAL DATA

We conducted further experiments to evaluate the recovery of ground-truth latent variables in synthetic data. Following the method in (Kapoor et al., 2024), we used the Lorenz attractor as the latent dynamics (3D latent variables). The visualizations of our decoded 3D trajectories presented in Fig. S7 confirmed that FDA effectively captured the neural dynamics.

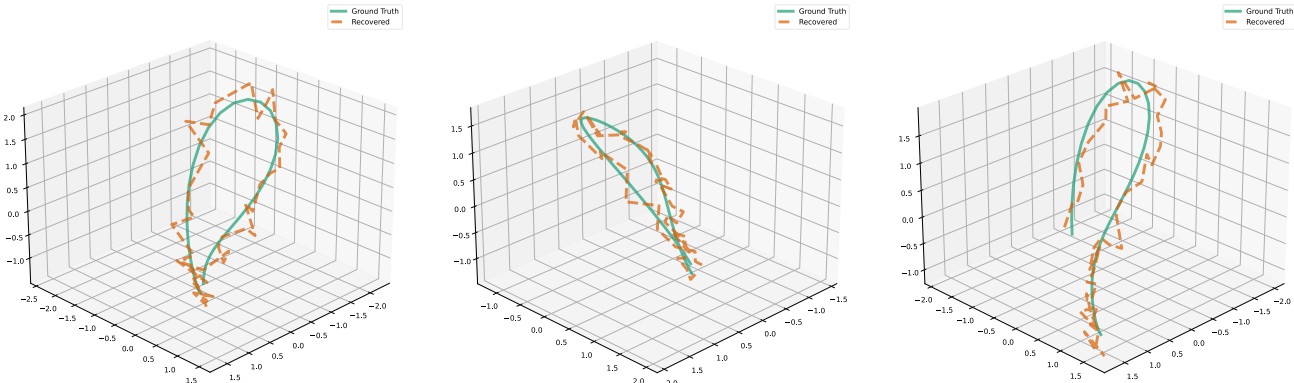

*Figure S7.* Ground truth (green, solid) and reconstructed 3D trajectories (orange, dashed) of the underlying Lorenz system from synthetic spiking data. FDA accurately recovers the latent dynamics.

## C.2. Ablation Study

### C.2.1. ABLATION STUDY ON DIFFERENT ALIGNMENT STRATEGIES

To evaluate the effectiveness of our alignment strategy, we compared FDA with several variants. FDA-t only extracted features using $f_\alpha$ and aligned them through MMD for decoding with a linear decoder. FDA-g used an adversarial approach via Cycle-GAN to align $\mathbf{z}(1)$, while FDA-c applied MMD for aligning $\mathbf{c}$. The average $R^2$ values of CO-C dataset are shown in Table S13.

Moreover, the $R^2$ curves for FDA-MMD and its variants are shown in Fig. S8(a). Additionally, as shown in Fig. S8(b), the negative log-likelihood (NLL) curves and their corresponding $R^2$ values, derived under various $r$ using FDA-MLA, are presented.

*Table S13.* Average cross-session $R^2$ scores (%) for CO-C. FDA-t only extracted features using $f_\alpha$ and aligned them through MMD for decoding with a linear decoder. FDA-g used an adversarial approach via Cycle-GAN to align $\mathbf{z}(1)$, while FDA-c applied MMD for aligning $\mathbf{c}$.

| Data | Target Ratio | FDA-t | FDA-g | FDA-c | **FDA-MLA** | **FDA-MMD** |
|------|------|------|------|------|------|------|
| CO-C | 0.02 | $-0.33_{\pm 0.29}$ | $13.19_{\pm 9.06}$ | $18.25_{\pm 7.30}$ | $16.39_{\pm 6.30}$ | $13.84_{\pm 5.41}$ |
| | 0.03 | $-0.30_{\pm 0.34}$ | $13.07_{\pm 9.06}$ | $18.49_{\pm 7.38}$ | $17.08_{\pm 6.53}$ | $13.93_{\pm 4.79}$ |
| | 0.04 | $-0.32_{\pm 0.28}$ | $13.06_{\pm 8.89}$ | $18.64_{\pm 7.43}$ | $17.27_{\pm 6.58}$ | $13.94_{\pm 5.64}$ |
| | 0.06 | $-0.23_{\pm 0.25}$ | $13.19_{\pm 8.94}$ | $18.60_{\pm 7.10}$ | $17.41_{\pm 6.66}$ | $13.82_{\pm 5.45}$ |

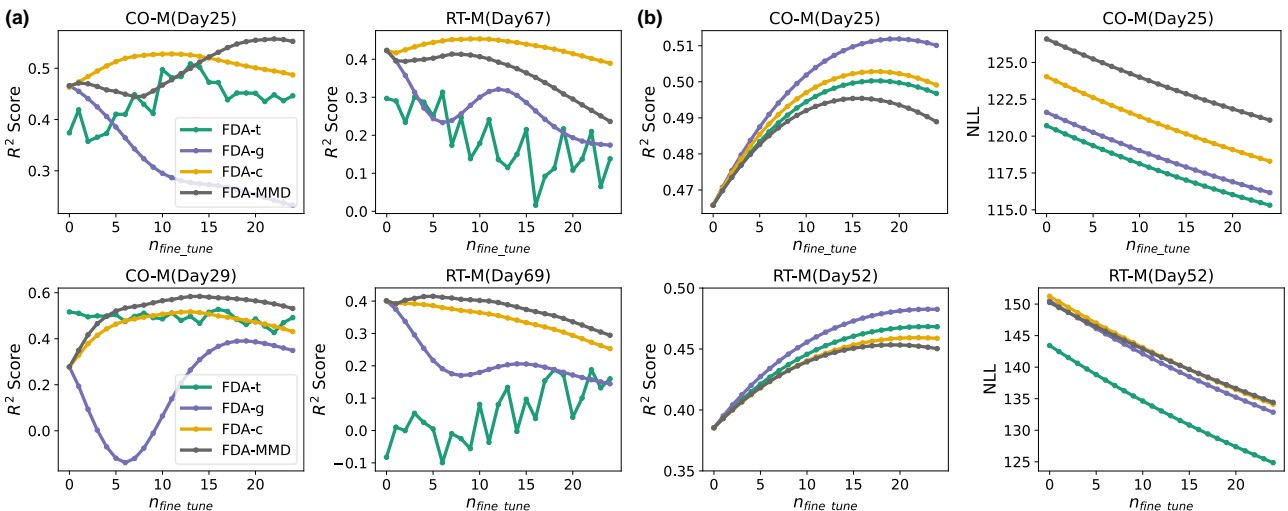

*Figure S8.* (a) $R^2$ curves for FDA-t, FDA-g, FDA-c, and FDA-MMD are shown on CO-M (Day25, Day29) and RT-M (Day67, Day69) with $r$ being 0.02. FDA-t only extracted features using $f_\alpha$ and aligned them through MMD for decoding with a linear decoder. FDA-g used an adversarial approach via Cycle-GAN to align $\mathbf{z}(1)$, while FDA-c applied MMD for aligning $\mathbf{c}$. (b) Curves for $R^2$ (Left) and the corresponding negative log likelihood (NLL) (Right) on CO-M (Day25) and RT-M (Day52), obtained by FDA-MLA, are visualized under distinct target ratios $r$.

To investigate the observed decline in $R^2$ with more finetuning samples as shown in Fig. 5(b), we further analyzed the average $R^2$ scores of FDA-MMD across multiple random seeds under varying training ratios $r$. The detailed results for each target session of the CO-M and RT-M datasets are provided in Table S14 and Table S15 below. A general trend of increasing $R^2$ with larger fine-tuning sample sizes was observed across most sessions.

*Table S14.* Average $R^2$ scores (%) across sessions for FDA-MMD on the CO-M dataset under varying $r$ values.

| $r$ | Day 8 | Day 14 | Day 15 | Day 22 | Day 24 | Day 25 | Day 28 | Day 29 | Day 31 | Day 32 |
|---|---|---|---|---|---|---|---|---|---|---|
| 0.02 | $45.23_{\pm4.44}$ | $55.90_{\pm3.17}$ | $49.55_{\pm3.41}$ | $27.35_{\pm7.34}$ | $51.28_{\pm2.53}$ | $36.79_{\pm4.12}$ | $54.87_{\pm4.40}$ | $41.26_{\pm5.70}$ | $57.10_{\pm3.24}$ | $44.66_{\pm4.41}$ |
| 0.03 | $44.47_{\pm3.31}$ | $59.18_{\pm4.24}$ | $52.90_{\pm4.62}$ | $40.58_{\pm4.98}$ | $55.94_{\pm1.78}$ | $41.10_{\pm2.95}$ | $57.93_{\pm4.03}$ | $39.56_{\pm6.33}$ | $59.15_{\pm1.77}$ | $48.08_{\pm2.90}$ |
| 0.04 | $46.68_{\pm2.44}$ | $60.35_{\pm5.54}$ | $53.18_{\pm5.23}$ | $42.89_{\pm4.10}$ | $59.48_{\pm2.23}$ | $45.84_{\pm2.40}$ | $59.97_{\pm2.62}$ | $42.66_{\pm5.25}$ | $61.03_{\pm1.72}$ | $49.80_{\pm3.28}$ |
| 0.06 | $49.96_{\pm3.43}$ | $60.48_{\pm5.33}$ | $52.53_{\pm5.18}$ | $43.19_{\pm3.64}$ | $59.19_{\pm2.65}$ | $49.29_{\pm3.97}$ | $61.25_{\pm2.38}$ | $45.06_{\pm3.81}$ | $63.31_{\pm2.97}$ | $51.27_{\pm2.66}$ |

*Table S15.* Average $R^2$ scores (%) across sessions for FDA-MMD on the RT-M dataset under varying $r$ values.

| $r$ | Day 1 | Day 38 | Day 39 | Day 40 | Day 52 | Day 53 | Day 67 | Day 69 | Day 77 | Day 79 |
|---|---|---|---|---|---|---|---|---|---|---|
| 0.02 | $74.32_{\pm2.25}$ | $55.39_{\pm2.80}$ | $40.44_{\pm7.31}$ | $39.85_{\pm3.27}$ | $44.99_{\pm4.96}$ | $50.03_{\pm4.44}$ | $50.29_{\pm5.07}$ | $39.19_{\pm4.07}$ | $16.67_{\pm9.32}$ | $38.99_{\pm5.70}$ |
| 0.03 | $73.86_{\pm3.20}$ | $58.12_{\pm2.75}$ | $41.61_{\pm5.91}$ | $41.88_{\pm3.17}$ | $44.87_{\pm5.25}$ | $52.17_{\pm0.71}$ | $51.08_{\pm6.30}$ | $43.40_{\pm3.84}$ | $20.51_{\pm8.44}$ | $41.95_{\pm5.26}$ |
| 0.04 | $74.57_{\pm2.45}$ | $58.79_{\pm2.71}$ | $41.39_{\pm6.34}$ | $42.20_{\pm4.26}$ | $45.09_{\pm5.13}$ | $53.39_{\pm1.56}$ | $51.27_{\pm6.91}$ | $45.08_{\pm3.98}$ | $22.68_{\pm7.64}$ | $41.69_{\pm5.51}$ |
| 0.06 | $74.98_{\pm1.93}$ | $58.97_{\pm1.54}$ | $43.82_{\pm6.86}$ | $43.50_{\pm4.54}$ | $45.03_{\pm5.34}$ | $53.76_{\pm1.35}$ | $52.65_{\pm5.39}$ | $44.26_{\pm4.33}$ | $29.05_{\pm7.01}$ | $48.94_{\pm5.36}$ |

In addition, we observed that the number of valid channels differs between the CO-M sessions (95 for the source and 96 for the target), whereas RT-M sessions remain consistent. This suggests less neuronal overlap with the source session in the CO-M dataset, resulting in worse decoding performance as presented in Fig. S8(a).

### C.2.2. ABLATION STUDY ON MAIN COMPONENTS

The average $R^2$ for each target session achieved by FDA and its variants based on main components is shown in Fig. S9. We employed FDA-v and FDA-p as variants utilizing VP and GVP flow paths, respectively. FDA-a and FDA-m employ transformers with temporal correlation attention and MLPs as conditional feature extractors, respectively. FDA-re includes an additional reconstruction term to regularize the learned neural representations.

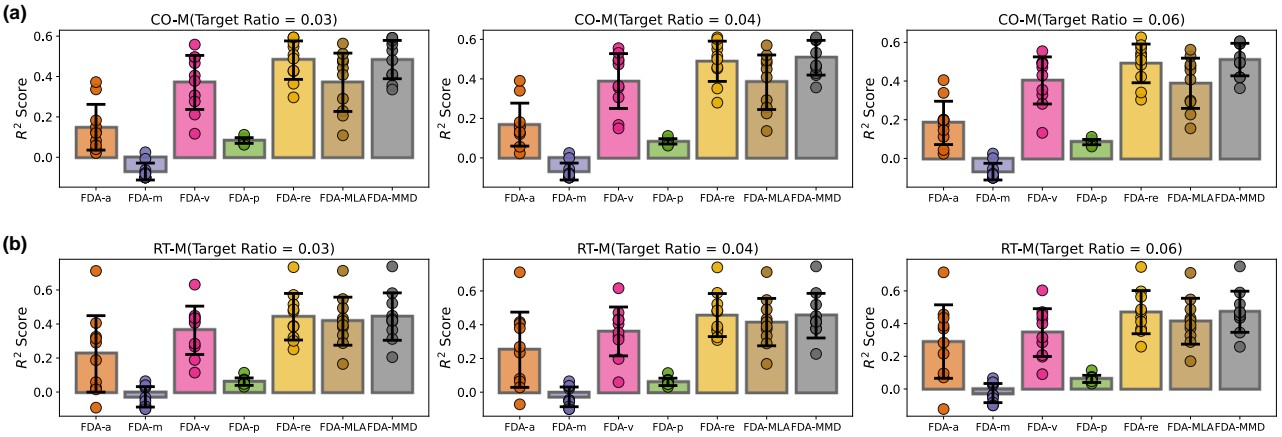

*Figure S9.* Average $R^2$ scores across each target session, achieved by FDA-a, FDA-m, FDA-v, FDA-p, FDA-re, FDA-MLA, and FDA-MMD, are displayed on CO-M (a) and RT-M (b) datasets with $r$ being 0.03, 0.04, and 0.06. Dots with different colors represent $R^2$ values for individual sessions. We employed FDA-v and FDA-p as variants utilizing VP and GVP flow paths, respectively. FDA-a and FDA-m employ transformers with temporal correlation attention and MLPs as conditional feature extractors, respectively. FDA-re includes an additional reconstruction term to regularize the learned neural representations.

## C.3. Hyper-parameter Sensitivity Analysis

The main hyper-parameters of our FDA method include the signal window size ($w$), the dimensions of conditional features and latent states($k_{c,z}$), and the number of euler sampling steps $n_{euler}$ when the target ratio $r$ equals 0.02. For convenience, we set $k_c$ and $k_z$ to be the same. The results of their sensitivity analysis using FDA-MMD on CO-M, and RT-M datasets are shown in Table S16, Table S17, and Table S18.

Table S16. Average $R^2$ scores for different datasets with varying $w$.

| $k_c$ | 4 | 5/6 | 7 | 8 |
|---|---|---|---|---|
| CO-M | $43.91_{\pm4.68}$ | $45.59_{\pm5.15}$ | $48.38_{\pm4.98}$ | $\mathbf{49.07}_{\pm5.11}$ |
| RT-M | $40.77_{\pm5.46}$ | $42.08_{\pm6.31}$ | $40.54_{\pm7.74}$ | $\mathbf{46.73}_{\pm3.83}$ |

Table S17. Average $R^2$ scores for different datasets with varying $k_c$.

| $k_c$ | 24 | 32 | 48 | 72 |
|---|---|---|---|---|
| CO-M | $\mathbf{48.00}_{\pm5.68}$ | $45.59_{\pm5.15}$ | $45.63_{\pm4.77}$ | $45.03_{\pm4.84}$ |
| RT-M | $\mathbf{44.02}_{\pm5.01}$ | $42.08_{\pm6.31}$ | $39.48_{\pm5.51}$ | $43.91_{\pm4.34}$ |

Table S18. Average $R^2$ scores for different datasets with varying $n_{euler}$.

| $n_{euler}$ | 1 | 2 | 4 | 10 |
|---|---|---|---|---|
| CO-M | $\mathbf{45.59}_{\pm5.15}$ | $45.32_{\pm5.14}$ | $43.19_{\pm5.34}$ | $41.71_{\pm5.37}$ |
| RT-M | $42.08_{\pm6.31}$ | $\mathbf{42.14}_{\pm6.17}$ | $40.33_{\pm6.12}$ | $38.99_{\pm6.23}$ |

We also conducted additional analyses on selected sessions from CO-M to investigate the desirable level of diversity in target trials. The diversity was categorized into three levels: small (1–2 distinct targets), medium (3–4 targets), and large (all distinct targets). The average $R^2$ for FDA-MMD is presented in Table S19. Our results indicate that a medium level of diversity achieves desirable performance, while too little diversity negatively impacts performance.

Table S19. Average $R^2$ scores (%) for FDA-MMD on CO-M under different levels of trial diversity

| Trial Diversity | Day14 | Day15 | Day28 | Day32 |
|---|---|---|---|---|
| Small | $58.93_{\pm1.10}$ | $45.86_{\pm1.81}$ | $50.90_{\pm1.70}$ | $43.70_{\pm2.60}$ |
| Medium | $57.01_{\pm1.80}$ | $57.26_{\pm1.20}$ | $55.46_{\pm1.00}$ | $49.19_{\pm1.49}$ |
| Large | $65.60_{\pm2.84}$ | $59.49_{\pm0.46}$ | $59.74_{\pm1.33}$ | $53.23_{\pm1.09}$ |

