# OpenReview forum: "Flow Matching for Few-Trial Neural Adaptation with Stable Latent Dynamics"
_ICML.cc/2025/Conference — ICML 2025 poster_

### Official Review · Reviewer_pQPy · 2025-03-13

**Overall Recommendation:** 4

**Summary:**

Neural representational drift is a well-known problem in brain computer interfaces (BCIs) that make it difficult to re-use a decoder over multiple days without an additional recalibration step. Importantly, the amount of data available each day for recalibration is often small. Prior work has explored ways of aligning neural activity across days to aid this process, but they often require more data than would be available in a real-world deployment. This paper argues that this is partially due to assumptions made on the latent variables and choice of divergences, which may break down in the low-data regime, leading to poor transfer between days and training instabilities. The authors propose Flow-Based Distribution Alignment (FDA), which leverages flow matching for few-trial neural adaptation. The choice of a flow matching-based method is motivated by the fact that it places minimal assumptions on the distributions of neural representations while enabling direct likelihood maximization. This second property also enables source-free alignment, which enables calibration without behavioral labels. They prove that their flow-based transformation is stable in terms of Lyapunov exponents, which they claim aids with transfer. They show that FDA improves performance over appropriate baselines on motor cortex datasets.

**Claims And Evidence:**

Overall, the paper did a good job supporting its claims with a thorough and extensive evaluation. However, it is still unclear to me why the stability of the flow is made out to be so critical for transfer. The baselines clearly do have larger Lyapunov exponents, but that doesn't prove that this is why they transfer poorly. I think this claim deserves a more thorough ablation. Either that, or it should not be so heavily focused on in the paper.

**Essential References Not Discussed:**

There are not essential references missing from this paper as far as I'm aware of the literature.

**Experimental Designs Or Analyses:**

I looked over the experimental design of the main results and ablations and found them to be sound.

**Methods And Evaluation Criteria:**

The proposed methods and evaluation criteria (both benchmarks and metrics) make sense for this problem setting.

**Other Comments Or Suggestions:**

I don't think that presenting the results in a table, as in Table 1, is the most effective way of communicating them in this case. While this is not uncommon in ML papers, I find this table to be too busy to parse. I'd suggest something more like a box or violin plot may help. Maybe with just the most important results, and then the rest could be moved to the supplementary material. This is not critical, but just a minor comment.

**Other Strengths And Weaknesses:**

The paper is well written and does a good job supporting most of its claims. It is a novel approach to neural representation alignment for BCI decoders, appears to have better few-shot transfer, and can do source-free alignment. Some of the technical innovations may also have broader application in other domains.

**Questions For Authors:**

Is there an easy way to ablate the stability property of FDA and show that when those assumptions are violated, it transfers worse in some way?

**Relation To Broader Scientific Literature:**

This paper provides clear arguments as to why their method should work better in the data-limited regime and empirically show it through extensive evaluations with state-of-the-art methods. The authors do a good job discussing related work and situating their paper in it. While the innovations in this paper are more specific to the application, there are definitely aspects which are relevant to the broader scientific community. Pre-training on large datasets and transferring with selectively re-training a subset of the model weights on new data is a common paradigm across machine learning. This paper makes important claims about how the choices of distribution of latent variables can have a substantial impact on this transferability. On a more low-level, their choice of linear interpolation for the flow path allowed them to use a one-step Euler method, simplifying their source-free alignment method. This flavor of technique may have utility in more settings than just neural representation alignment. They also prove that their flow is stable under certain assumptions and regularizations, which to my knowledge, is novel to this work.

**Theoretical Claims:**

I skimmed the proof of dynamical stability in appendix A2 and nothing popped out to me as incorrect.

---

> ### Author Rebuttal · Authors · 2025-04-01
>
> We sincerely thank you for your careful review and recognition of our work. Below, we provide a point-by-point reply to your concerns. Due to the limit, all figures prefixed with 'R' below are available in the external link [https://drive.google.com/file/d/129vv370SF4RLanLj92-lzh_vMmkeDCve/view?pli=1].
>
> ### Claims And Evidence:
>
> - About why the stability of the flow is made out to be so critical for transfer and a more thorough ablation
>
> Thanks for raising the unclear point. The stability of flow models enhances zero-shot performance, making it effective for few-trial adaptation. Specifically, this stability regulates feature deviations, preserving essential semantic information across domains. As illustrated in Fig.R4, it ensures that latent factors with similar labels flow toward consistent semantic representations, even under input drifts in target signals. Empirical validation through zero-shot transfer performance (original Table S8 on Page 18) further demonstrates the benefits of this stability.
>
> Additionally, we conducted a more thorough ablation study to further validate this effect. Since stability is maintained by activation functions and scale coefficients, we ablated these two components (FDA-al and FDA-sc, respectively) to violate the assumptions of stability. As shown in Fig.R5, the distribution of maximum Lyapunov exponents (MLE) for FDA-al and FDA-sc indicates that both variants frequently exhibit positive MLEs, signifying instability. Furthermore, the corresponding results for zero-shot and few-trial performance, evaluated using MMD-based alignment on the CO-M and RT-M datasets, are summarized in the table below. Consistent with their reduced stability, FDA-al and FDA-sc demonstrated substantially degraded transfer performance, which is comparable to the baselines. This result suggests that instability is the factor contributing to their poor transfer ability.
>
> #### Comparison of average $R^2$ scores (%) across sessions for FDA-al, FDA-sc, and FDA-MMD on the CO-M, and RT-M datasets ($r$ = 0, 0.02).
>
> | Data | $r$ | FDA-al | FDA-sc | FDA-MMD |
> |:---------:|:-----------:|:-------------------:|:----------------------:|:--------------------:|
> | CO-M  | 0        | -9.34 ± 9.57        | -18.20 ± 17.53         | **16.23 ± 9.43**         |
> |           | 0.02        | 14.51 ± 16.37       | 13.35 ± 19.01          | **45.59 ± 5.15**         |
> | RT-M  | 0       | 1.23 ± 4.76         | 1.78 ± 3.89            | **38.15 ± 8.21**         |
> |           | 0.02        | 16.99 ± 11.75       | 20.46 ± 11.77          | **42.08 ± 6.31**         |
>
> ### Other Comments Or Suggestions:
>
> - I don't think that presenting the results in a table, as in Table 1, is the most effective way of communicating them in this case. While this is not uncommon in ML papers, I find this table to be too busy to parse. I'd suggest something more like a box or violin plot may help. Maybe with just the most important results, and then the rest could be moved to the supplementary material.
>
> Thanks a lot for the valuable suggestion. We will improve the presentation of our results by using clearer violin plots and relocating some results to the supplementary material in our revision.
>
> ### Questions For Authors:
>
> - About the easy way to ablate the stability property of FDA and show that it transfers worse in some way when those assumptions are violated
>
> Yes, we found that FDA exhibited significantly worse zero-shot and few-trial transfer performance when these assumptions were violated. Specifically, since stability is governed by activation functions and scale coefficients, we ablated these two components (FDA-al and FDA-sc, respectively) to affect the stability property of FDA. More detailed results are provided above in the above response.
>
> We sincerely hope that these responses may address your concerns. We believe that our novel FDA framework will be of significant interest to the ICML community, given its potential impact on few-trial neural alignment and real-world BCI reliability.
> Could you please consider raising the scores? We look forward to your further feedback. Thank you in advance.

---

> > ### Comment · Reviewer_pQPy · 2025-04-03
> >
> > I want to thank the authors for performing the requested ablations! Although it is not surprising, I felt it was an important aspect to validate empirically.

---

### Official Review · Reviewer_CxsD · 2025-03-14

**Overall Recommendation:** 3

**Summary:**

This paper proposes Flow-based Distributional Alignment (FDA), a few-shot alignment or adaptation method for neural signals across days, using flow matching. While neural activity adaptation methods in general struggle to maintain stable performance across multiple days, the authors claim that FDA would perform better due to theoretical guarantees on the stability of representations derived from FDA. That is, due to the highly negative maximum Lyapunov exponent of latent states through learning, the authors argue that FDA-finetuning results in latent representations that converge to a stable fixed point or manifold/attractor. The authors carry out comprehensive experiments on a motor cortical dataset containing data from 2 monkeys, and 2 tasks (centre-out and random target reaches) to demonstrate the performance of their model in comparison to baselines. They also carry out ablation studies to gauge the importance of each part of their method.

### Update after rebuttal

My main outstanding concerns are summarised in my response to the authors' rebuttal. Finally, the NDT-2 0-shot results here don't seem consistent with Fig. 5 from the original NDT-2 paper, showing good 0-shot and few-shot performance (even for real-time control). In the absence of more details on the authors' reproduction of NDT-2 and 0-shot setup, I'm unable to evaluate this result completely. So overall, I will still retain my original score.

**Claims And Evidence:**

The main claim of the paper is that the proposed method, FDA, outperforms baseline methods in few-shot adaptation to new days. Based on the experiments in the paper, I mostly think that there is sufficient evidence for this claim. The claim about novelty is valid – to my knowledge, this is a novel application of flow matching to neural activity adaptation. The sample and compute efficiency claims also seem to hold in light of experimental results.

**Essential References Not Discussed:**

I think relevant and related work is adequately discussed in the paper.

**Experimental Designs Or Analyses:**

Yes, I believe the experiments carried out and the analyses are sound. The authors have correctly validated their claims of superior few-shot performance on cross-day neural decoding. The claim about stability of latent dynamics (through training) is also validated empirically through the Lyapunov exponent analysis. Several ablations have been carried out, validating the necessity and utility of various model components.

**Methods And Evaluation Criteria:**

Yes, the proposed methods, evaluation datasets, and metrics (decoding $R^2$) make sense for the application at hand. Quantifying stability of dynamics through the maximum Lyapunov exponent is also a standard practice.

**Other Comments Or Suggestions:**

One suggestion to the authors would be to attempt to make the writing clearer and accessible to a BCI audience by retaining the most important details and experiments in the main text and relegating some of the others to the appendix. Given the number of ablations, it gets confusing to remember and map the abbreviated variants of FDA to particular components being ablated, especially when looking at figures. Some of the table captions, such as that of Table S9, could be made more descriptive. I had to go back to the main text to understand what I was looking at. Also, some figures lack certain axes labels such as Figure S4, which makes it hard to interpret them.

**Other Strengths And Weaknesses:**

Apart from points made previously, here are additional strengths and weaknesses:

**Strengths:**
* The ability to adapt few-shot using even just 5 trials is a great advantage.
* The method does not require target trials to be labelled, so learning the adaptation is unsupervised.

**Weaknesses:**
* The paper is very dense and so is a lot to take in, but it also lacks clarity in certain places (see Questions section).
* Some cited works such as Azabou et al. (2023; POYO), Ye et al. (2024; NDT-2), etc. were not compared against. I think it should be possible to compare against models like NDT-2 where the source code and weights are available (https://github.com/joel99/context_general_bci).
* Related to the above, there is a claim in Section 2 that Ye at al. (2023; NDT-2) and Zhang et al. (2024; MtM) are capable of supervised alignment. I do not quite understand this point: NDT-2 and MtM are trained in a self-supervised manner, and only the readout training is supervised. Furthermore, alignment isn't really necessary in either case because they are large-scale models trained on multiple sessions/tasks/datasets to begin with. Could the authors clarify what they mean here and perhaps modify the text here? It is true that Azabou et al. (POYO; 2023) is amenable to supervised alignment.
* Other BCI tasks such as speech and handwriting decoding were not considered.
* As acknowledged by the authors, cross-task and cross-subject adaptation was not studied.

**Questions For Authors:**

Apart from my comments above, here are some questions:
* In some of the plots, such as Figure 4(b) and Figure S6, why does the $R^2$ drop as the number of samples for finetuning increases?
* How do the authors decide the window size $w$ for the context windows?
* In some cases, performance on the random target task is better than performance on centre-out. Do the authors have any intuition for why that might be the case? In general, decoding in RT tasks is more difficult than CO, and RT is a harder task for subjects to do as well.
* The authors randomise the selection of few-shot finetuning trials, but could they comment on what level of diversity in the trials is desirable? For example, if all the finetuning trials sampled were associated with just one of the centre-out targets, I would expect the performance to be low – although in the real-world setting, one could argue that the experimenter can collect calibration data with some diversity.
* I recall reading that the latent state was read out using a linear decoder somewhere in the paper, but could the authors clarify if my understanding is correct? This could be made clearer.
* What are the details of the hardware used for the computational efficiency benchmarking? The time in seconds seems very low if that's the training time. Is it the total time taken or the time taken for 1 step/epoch?
* Can the authors comment on the computational efficiency in terms of inference time? This is important given that the causal formulation allows for real-time inference.

**Relation To Broader Scientific Literature:**

This paper makes a contribution towards improving neural activity alignment methods, which is one class of methods meant to improve the performance of BCI decoders across days or even subjects (although the latter is not explored here and acknowledged as a limitation). Existing methods of this class often require a large number of trials to adapt neural activity on subsequent days to the source domain (usually day 0), however the proposed method is able to achieve good results with very few trials. It is also, to my knowledge, one of the first works applying flow matching to this setting. It is worth noting that this class of adaptation-based methods often end up yielding poorer performance than large-scale deep learning methods that have been proposed recently, however the asset of the proposed method is its ability to generalise in the setting with extremely few and unlabelled trials for finetuning.

**Theoretical Claims:**

The theoretical result and the proof seem to make sense, I briefly went through the full proof in the appendix but did not read it in full detail.

---

> ### Author Rebuttal · Authors · 2025-04-01
>
> We sincerely thank you for your careful review and recognition of our work. Below, we provide a point-by-point reply to your concerns.
>
> - Additional comparison with NDT-2
>
> We conducted an additional comparison against NDT-2 by pre-training it on a single session with supervised readout training and evaluating its zero-shot performance without alignment. The average $R^2$ scores on the CO-C and CO-M datasets are presented in the table below. While NDT-2 achieved high decoding performance in the pre-trained session, its zero-shot performance degraded significantly. This decline may be attributed to its reliance on larger datasets to achieve robust zero-shot transfer.
>
> #### Average $R^2$ scores (%) of NDT-2
>
> | Data  | intra-session | inter-session |
> |:------: |:-----------------:|:-----------------:|
> | CO-C | 83.85 ± 1.49      | -28.59 ± 0.49     |
> | CO-M | 82.09 ± 1.06      | -35.69 ± 2.06     |
>
> - About the supervised alignment by NDT-2 and MtM
>
> Thank you for the valuable comment. We agree that alignment of large-scale models isn't necessary in the cases we have presented in this study.
>
> What we meant is that the finetuning of MtM and NDT-2 can begin with self-supervised techniques, such as masked reconstruction. Then, for different downstream tasks, specific decoders are trained using only a few target labels, while all other weights remain fixed. This supervised alignment enables rapid transfer to tasks that were not encountered during pre-training.
>
> - About other BCI tasks and cross-task/subject adaptation
>
> Thanks for the comment. We agree that this presents interesting directions for further study.
>
> ### Questions:
>
> - About the drop of $R^2$ with more finetuning samples
>
> The curve in Fig.4(b) and Fig.S6 is based on a single random run, and the observed decrease may be due to the small trial gap (~2 trials between adjacent points). Moreover, the overall trend shows an increase in $R^2$ as the number of finetuning samples grows, as demonstrated in Fig.3(c) on Page 6.
>
> - About the choice of window size w
>
> Thanks. We conducted a grid search to determine the appropriate context window size $w$. As shown in the Table S11 on Page 21, balancing performance and computational efficiency, we selected 5 as the default size for the CO-M and RT-M.
>
> - About better $R^2$ on RT than CO tasks
>
> The better performance on RT tasks may be attributed to differences in signal quality between subjects. For the same subject (Monkey M), performance on CO tasks is better than on RT tasks, as shown in the original Fig.3(b) on Page 6.
>
> - About the desirable level of diversity in trials
>
> We conducted additional analyses on selected sessions of CO-M. The diversity was categorized into three levels: small (1–2 distinct targets), medium (3–4 targets), and large (all distinct targets). The average $R^2$ for FDA-MMD is presented in the tables below. Our results indicate that a medium level of diversity achieves desirable performance, while too little diversity negatively impacts performance.
>
> #### Average $R^2$ scores (%) for FDA-MMD on CO-M
>
> | Trial Diversity   | Day14  | Day15  | Day28 | Day32 |
> |:------:|:------------------:|:------------------:|:------------------:|:------------------:|
> | Small  | 58.93 ± 1.10       | 45.86 ± 1.81       | 50.90 ± 1.70       | 43.70 ± 2.60       |
> | Medium | 57.01 ± 1.80       | 57.26 ± 1.20       | 55.46 ± 1.00       | 49.19 ± 1.49       |
> | Large  | 65.60 ± 2.84       | 59.49 ± 0.46       | 59.74 ± 1.33       | 53.23 ± 1.09       |
>
> - About the linear decoder
>
> The linear decoder was mentioned in Line 162 (right) on Page 3, and will be made clearer in the revision.
>
> - About details of hardware and reported computational efficiency
>
> The hardware used was NVIDIA GeForce RTX 3080 Ti (12GB). The training time reported in Table S9 on Page 20 summarizes the time per epoch for both pre-training and fine-tuning. A more detailed comparison of total time(s) is presented below.
>
> |     | ERDiff  | Cycle-GAN | NoMAD  | FDA-MLA | FDA-MMD |
> |:-------------------:|:-------:|:---------:|:------:|:-------:|:-------:|
> | Pre-training           | 2449.68 | -         | 135.34 | 98.95   | 98.95   |
> | Fine-tuning            | 1.21    | 11.90     | 76.83  | 2.32    | 2.38    |
>
> - About the efficiency of inference
>
> We conducted further analysis of FDA's inference time on an NVIDIA GeForce GTX 1080 Ti (11GB). As presented in the table below, average inference time for a single window is approximately 4 ms, making it suitable for real-time applications.
>
> #### Inference time of FDA (ms)
>
> | Data   | FDA-MLA | FDA-MMD |
> |:------:|:-------:|:-------:|
> | Avg | 3.90 | 3.97 |
>
> We hope that these responses may address your concerns. We believe that our novel FDA framework will be of significant interest to the ICML community, given its potential impact on few-trial neural alignment and real-world BCI reliability. Could you please consider raising the scores? We look forward to your further feedback. Thank you in advance.

---

> > ### Comment · Reviewer_CxsD · 2025-04-04
> >
> > I thank the authors for their response.
> >
> > > Additional comparison with NDT-2
> >
> > This is not really a fair comparison. The comparison should be done with a pre-trained NDT-2 model that is only fine-tuned few-shot to the target session using some data. The rationale behind this is that powerful pre-trained models are available publicly and can be fine-tuned for specific use cases with very little labelled data. The experiment done here misses this key ingredient and is against the reasoning behind large-scale decoding approaches.
> >
> > Considering POYO for example, a pre-trained model may be fine-tuned to new sessions few-shot using very little labelled data, and by only re-training embeddings for neuron/unit identity (unit re-identification pipeline in POYO, see some results [here](https://poyo-brain.github.io)). This still leads to good performance even on unseen animals.
> >
> > > About the drop of $R^2$ with more finetuning samples
> >
> > Shouldn't the authors be including error bars on these plots, and showing the (mean +/- std/SEM) improvement across several runs/seeds or even across days instead of for a particular day? I think this would be important for all figures (other ones seem to have error bars). If the same trend is still captured, then it is not simply due to randomness but a failure case of the method, where it doesn't work as effectively for a larger number of finetuning  samples – which would be important to understand.
> >
> > > About better $R^2$ on RT than CO tasks
> >
> > Yes, I understand that and have looked at Figure 3, however my point is mainly about Figure 4 in this case where there is about 0.1 higher $R^2$ for the RT task. I am curious if the authors could explain this in terms of number neurons in the recording or other metrics. I'm also curious why the plots are for CO-M Day 31 in (a) and Day 29 in (b), while RT remains the same (Day 52).
> >
> > I acknowledge the additional experiments on finetuning trial diversity and benchmarking – these would be good additions to the paper. I still maintain my positive opinion overall and lean towards acceptance, but I will retain my score.

---

> > > ### Author Response · Authors · 2025-04-07
> > >
> > > We are grateful for your further feedback and provide our responses as follows.
> > >
> > > - Additional comparison with NDT-2
> > >
> > > Thank you for the suggestion. For a fairer comparison, we utilized 47 sessions recorded from the motor cortex of two monkeys, available via the external link (https://zenodo.org/records/3854034), as well as datasets provided by the Neural Latents Benchmark (https://neurallatents.github.io/) for pre-training. Subsequently, supervised fine-tuning was conducted using 80% of the trials from a source session of the CO-M/RT-M datasets, followed by zero-shot evaluation on the remaining target sessions.
> > >
> > > The average $R^2$ scores are presented in the table below. While NDT-2 achieved higher performance in intra-session decoding on the source session, its performance degraded substantially during zero-shot evaluation. This suggests that NDT-2 may be more effective when fine-tuned with data from the target sessions.
> > >
> > > #### Average $R^2$ scores (%) of NDT-2
> > >
> > > | Data  | intra-session       | inter-session      |
> > > |:------|:-------------------:|:------------------:|
> > > | CO-C  | 89.82 ± 1.18        | -34.70 ± 0.32      |
> > > | CO-M  | 93.75 ± 1.51        | -52.31 ± 0.16      |
> > >
> > > - About the drop of $R^2$ with more finetuning samples
> > >
> > > We further analyzed the average $R^2$ scores of FDA-MMD across multiple random seeds for each target session under varying training ratios $r$ on the CO-M and RT-M datasets. The detailed results are provided in the tables below. A general trend of increasing $R^2$ with larger fine-tuning sample sizes was observed across most sessions.
> > >
> > > The drop in $R^2$ on Day 52 (RT-M) is attributed to FDA-MMD achieving similar performance regardless of the number of fine-tuning samples, making it an exceptional case. Additional results, showing the (mean +/- std/SEM) improvement, will be annotated in our original figures.
> > >
> > > #### Average $R^2$ scores (%) on CO-M dataset under varying $r$
> > > | $r$ | Day 8          | Day 14         | Day 15         | Day 22         | Day 24         | Day 25         | Day 28         | Day 29         | Day 31         | Day 32         |
> > > |:-:|:-:|:-:|:-:|:-:|:-:|:-:|:-:|:-:|:-:|:-:|
> > > | 0.02    | 45.23 ± 4.44   | 55.90 ± 3.17   | 49.55 ± 3.41   | 27.35 ± 7.34   | 51.28 ± 2.53   | 36.79 ± 4.12   | 54.87 ± 4.40   | 41.26 ± 5.70   | 57.10 ± 3.24   | 44.66 ± 4.41   |
> > > | 0.03    | 44.47 ± 3.31   | 59.18 ± 4.24   | 52.90 ± 4.62   | 40.58 ± 4.98   | 55.94 ± 1.78   | 41.10 ± 2.95   | 57.93 ± 4.03   | 39.56 ± 6.33   | 59.15 ± 1.77   | 48.08 ± 2.90   |
> > > | 0.04    | 46.68 ± 2.44   | 60.35 ± 5.54   | 53.18 ± 5.23   | 42.89 ± 4.10   | 59.48 ± 2.23   | 45.84 ± 2.40   | 59.97 ± 2.62   | 42.66 ± 5.25   | 61.03 ± 1.72   | 49.80 ± 3.28   |
> > > | 0.06    | 49.96 ± 3.43   | 60.48 ± 5.33   | 52.53 ± 5.18   | 43.19 ± 3.64   | 59.19 ± 2.65   | 49.29 ± 3.97   | 61.25 ± 2.38   | 45.06 ± 3.81   | 63.31 ± 2.97   | 51.27 ± 2.66   |
> > >
> > > #### Average $R^2$ scores (%) on RT-M dataset under varying $r$
> > >
> > > | $r$ | Day 1          | Day 38         | Day 39         | Day 40         | Day 52         | Day 53         | Day 67         | Day 69         | Day 77         | Day 79         |
> > > |:-:|:-:|:-:|:-:|:-:|:-:|:-:|:-:|:-:|:-:|:-:|
> > > | 0.02   | 74.32 ± 2.25   | 55.39 ± 2.80   | 40.44 ± 7.31   | 39.85 ± 3.27   | 44.99 ± 4.96   | 50.03 ± 4.44   | 50.29 ± 5.07   | 39.19 ± 4.07   | 16.67 ± 9.32   | 38.99 ± 5.70   |
> > > | 0.03    | 73.86 ± 3.20   | 58.12 ± 2.75   | 41.61 ± 5.91   | 41.88 ± 3.17   | 44.87 ± 5.25   | 52.17 ± 0.71   | 51.08 ± 6.30   | 43.40 ± 3.84   | 20.51 ± 8.44   | 41.95 ± 5.26   |
> > > | 0.04   | 74.57 ± 2.45   | 58.79 ± 2.71   | 41.39 ± 6.34   | 42.20 ± 4.26   | 45.09 ± 5.13   | 53.39 ± 1.56   | 51.27 ± 6.91   | 45.08 ± 3.98   | 22.68 ± 7.64   | 41.69 ± 5.51   |
> > > | 0.06   | 74.98 ± 1.93   | 58.97 ± 1.54   | 43.82 ± 6.86   | 43.50 ± 4.54   | 45.03 ± 5.34   | 53.76 ± 1.35   | 52.65 ± 5.39   | 44.26 ± 4.33   | 29.05 ± 7.01   | 48.94 ± 5.36   |
> > >
> > > - About better $R^2$ on RT than CO tasks
> > >
> > > We further analyzed the mutual information (MI) between spiking recordings from individual channels and calculated the maximum MI values across channels to assess the similarity between the source and target sessions.
> > > We found that Days 29 and 31 from CO-M achieved an average maximum MI of about 6e-4, which is significantly lower than the 2e-3 observed on Day 52 (RT-M).
> > >
> > > Additionally, we observed that the number of valid channels differs between the CO-M sessions (95 for the source and 96 for the target), whereas RT-M sessions remain consistent. This suggests less neuronal overlap with the source session, resulting in worse decoding performance. Moreover, relative figures for different days (Day 67 and Day 69) of the RT-M dataset are presented in Figure S6(a) on Page 20.
> > >
> > > Thanks once again for your valuable feedback. We look forward to any additional questions you may have.

---

### Official Review · Reviewer_Uwcf · 2025-03-16

**Overall Recommendation:** 3

**Summary:**

The work propose to utilize Flow-based distribution alignment (FDA) to learn flexible neural representations with stable latent dynamics, and performs source free alignment with likelihood maximization. The author additionally performed theoretical analysis on the stability of latent dynamics, and performed experiments on multiple motor cortex datasets.

**Claims And Evidence:**

The claim that BCIs trained on one day usually obtain degraded performance on other days due to the nonstationary property
of neural signals makes sense. Actually the signal pattern could be quite different even across different sessions within the same day.

**Essential References Not Discussed:**

The work tackles the variability issue of brain signal decoding. While the work provided pretty nice review on the method side including neural representation alignment and normalizing flows, I found the related works that specifically tackling the variability of brain signal decoding currently missing, including [1][2] etc.


[1] Distributionally Robust Cross Subject EEG Decoding, ECAI 2023
[2] UNCER: A framework for uncertainty estimation and reduction in neural decoding of EEG signals, Neurocomputing 2023

**Experimental Designs Or Analyses:**

The authors performed pretty detailed comparison with numerous existing baselines, on top of three real-world datasets. Additionally, ablation study including different alignment strategies and the contribution of each of the components are analyzed. Overall the experiments looks pretty solid. I would appreciate it if the authors could provide more intuitive visualization on the flow matching process.

**Methods And Evaluation Criteria:**

The method of using flow-based distribution alignment to tackle the variability in signal decoding looks pretty novel to me. However, stronger motivation is needed on the proposed approach. In addition to the general description that FDA fits well to the problem, the readers would want to know what are the exact benefits that the mechanism could bring to tackling the problem, and what functionality makes the method specifically fit to the problem.

**Other Comments Or Suggestions:**

More annotations are needed in Fig. 2 for readers to properly understand the mechanism of the approach. For example, what does X^S and X^T respectively stands for. Similarly for C^S and C^T. And what is the relationship between C^S and v_{\theta}?

More explanation is needed on why target distribution p1 representing the desired neural representation could be defined by a random variable z^S(1)?

**Other Strengths And Weaknesses:**

N/A

**Questions For Authors:**

Please see above comments

**Relation To Broader Scientific Literature:**

The method could be useful in numerous downstream applications utilizing BCI systems for rehabilitation and other healthcare purposes.

**Theoretical Claims:**

The theoretical analysis on stability of latent dynamics is provided in the manuscript, I didn't check the details on the correctness.

---

> ### Author Rebuttal · Authors · 2025-04-01
>
> We sincerely thank you for your careful review and recognition of our work. Below, we provide a point-by-point reply to your concerns. Due to the limit, all figures prefixed with 'R' below are available in the external link[https://drive.google.com/file/d/129vv370SF4RLanLj92-lzh_vMmkeDCve/view?pli=1].
>
> ### Methods And Evaluation Criteria
>
> - About the motivation
>
> Our FDA successfully addresses the issues of significant degradation in zero-shot performance and instability in few-trial alignment, achieving stable latent feature extraction and reliable alignment with few trials.
>
> On one hand, flow-based learning regulates feature deviations through stable latent dynamics, preserving essential semantic information from the source domain. This mechanism enforces bounded variations in latent features under input target perturbations, preserving semantic information in neural representations. Empirical validation of zero-shot transfer performance (original Table S8 on Page 18) highlights the advantage of this stability, further enhancing efficient few-shot adaptation.
>
> On the other hand, by leveraging the unique properties of flows, FDA guarantees stable few-trial fine-tuning. The use of explicit log-probability objectives and flexible latent-space modeling enables stable parameter gradients, effectively preventing the catastrophic overfitting typically seen in few-trial adaptation.
>
> ### Experimental Designs Or Analyses
>
> - About more intuitive visualization on the flow matching process
>
> Thank you for the suggestion. We have provided a more intuitive visualization of the flow matching process in Fig.R4. As shown in this figure, our flow matching process learns latent variables from neural signals in a coarse-to-fine manner, differing from conventional one-step extraction.
>
> The flow transforms noisy variables $\mathbf{z}(0)$ into neural representations $\mathbf{z}(1)$, guided by conditional features derived from neural signals. Simultaneously, the corresponding prior distribution $p_0$ is transformed into the target distribution $p_1$. The stable latent dynamics of the learning process ensure that latent factors with similar labels flow toward consistent neural representations, even when guided by shifted neural signals.
>
> ### Essential References Not Discussed
>
> - About the missing related works that specifically tackling the variability of brain signal decoding
>
> Thanks a lot for the suggestion. We will include more related works that specifically tackle the variability of brain signal decoding in the revision.
>
> ### Other Comments Or Suggestions:
>
> - About more annotations in Fig. 2. For example, what does X^S and X^T respectively stands for. Similarly for C^S and C^T. And what is the relationship between C^S and v_{\theta}?
>
> Thank you for the valuable comment. We have added more annotations, and the revised version is provided as Fig.R2. Here, $\mathbf{x}^S$ and $\mathbf{x}^T$ represent input tokens derived from short-term windows of the source and target domains, with each token corresponding to spikes from a single channel. Additionally, $\mathbf{c}^S$ and $\mathbf{c}^T$ denote the output conditional features of the transformer-based $f_{\alpha}$, which are utilized to guide the flow of $\mathbf{z}$.
>
> As illustrated in Fig.R4, we parameterize the guided velocity field of $\mathbf{z}$ as $v_{\theta}$, which determines the velocity and direction of $\mathbf{z}$. $\mathbf{c}^S$ further serves as the input to the neural network responsible for predicting $v_{\theta}$.
>
> - About more explanations on why target distribution p1 representing the desired neural representation could be defined by a random variable z^S(1)?
>
> Thank you for raising this point. To achieve optimal decoding performance, the target distribution is preferable to be defined by a random variable, which can then be transformed into the ground-truth labels through the decoders. When weights of the linear decoder are fixed, the desired $\mathbf{z}^S(1)$ can be obtained via a linear transformation using the inverse of the decoder's weight matrix. The neural representation thus defined can subsequently be transformed into the desired labels through the decoder. Related explanation will be included in the revision.
>
> We sincerely hope that these responses may address your concerns. We believe that our novel FDA framework will be of significant interest to the ICML community, given its potential impact on few-trial neural alignment and real-world BCI reliability.
> Could you please consider raising the scores? We look forward to your further feedback. Thank you in advance.

---

### Official Review · Reviewer_3LVu · 2025-03-16

**Overall Recommendation:** 3

**Summary:**

The paper introduces a new approach for learning and aligning neural representations across multiple sessions to link neural activity with behavioral actions. Particularly, the authors present a neural decoder that aligns recordings from different sessions (e.g., over multiple days)
 based on flow matching in latent space. This method is particularly useful when only a few trials are available for alignment, a common issue in brain-computer interfaces. The paper also provides a theoretical proof for the stability of the approach using Lyapunov exponents. The authors demonstrate the effectiveness of their method on various monkey neural recording datasets and benchmark it against existing approaches.

**Claims And Evidence:**

Yes

**Essential References Not Discussed:**

No, I think that overall the related work section was written well

**Experimental Designs Or Analyses:**

Yes, please also see the 2nd point in strengths and weaknesses.

**Methods And Evaluation Criteria:**

Yes, please also see the 2nd point in strengths and weaknesses.

**Other Comments Or Suggestions:**

1) I miss a comma before “resulting” in line 11 (right).
2) In line 111 (left), you mention that the labels are in $ \mathbb{R}^d $; it would be helpful to clarify from the beginning that labels can be multi-dimensional (and also explain what $ d $ refers to).
3) The notation for $ z^{S}(\tau) $ is a bit confusing since for $ x $, the input in brackets was the channel number, and the subscript referred to time. Why not be consistent by using the subscript for time here as well?
4) In line 181, you mention the scale coefficient, but I cannot see where it is used (at least in the main text). Are you sure you defined that?
5) In line 191 and Equations 5 and 7, is it all $ \ell_2 $ distance? Why not to clarify it is $\ell_2$ in the subscript to distinguish it from other norms (especially since you use other norms later)?
6) I believe lines 236-242 (left) would be better placed in the related work section.
7) It seems that the authors frame the motivation for the paper around BCIs; however, the model can also work on non-BCI neural recordings (e.g., neuropixels), as demonstrated on some of the datasets. It is unclear to me why the authors chose to focus mainly on  BCIs when the model doesn’t seem to integrate unique BCI features (e.g., feedback).

**Other Strengths And Weaknesses:**

## Strengths:
 The paper is overall well-written and addresses a very important problem. The authors provide the assumptions, mathematical developments, and theory to support their claims and effectively motivate the need for their model. I also appreciate the supplementary material, which further supports the paper with helpful additional explanations.

## Weaknesses:
 My main two concerns are: 1) Model interpretability, and 2) Lack of evaluation on synthetic data. Particularly:

1) The model is very complex, as it includes the tuning of multiple networks, which are themselves not very interpretable. In other words, the fundamental components of the model may be very hard to understand or explain in relation to neural dynamics. Given that the paper focuses on BCIs that could ultimately be used in clinical settings, it’s hard to believe clinicians would trust a model that is not easily interpretable. Hence, I wonder if there was any effort from the authors to interpret the components themselves (e.g., f_alpha) to provide some understanding of what features they emphasize, etc.


2) The authors demonstrate their method only on real-world monkey data, but they do not show recovery of ground-truth latent variables in synthetic data, which I believe is crucial to show that the model can truly recover the real underlying components. Specifically, I am referring to a test beyond just evaluating R² of reconstruction—creating simulated data with real known z, generating “observations” from this z, and then showing that the model can recover Z. Without this, while the model provides good reconstruction as demonstrated in the results, we cannot be sure it can recover the real underlying latent dynamics.


3) How realistic is the assumption made on lines 196-197 regarding the z evolving linearly and monotonically between 0 and 1? I assume this choice is meant to regularize the model’s development of z, but some intuition for its implications (perhaps in the discussion) would be helpful.


I also have smaller concerns listed under “Other Comments or Suggestions".

**Questions For Authors:**

1. How do you define a session? Do different sessions necessarily come from the same subject, or can they involve different subjects with the model learning a unified representation for them? Is it always the same subset of neurons observed across sessions? I think a better clarification of how sessions can vary (e.g., subject identity, observed neurons) would be helpful. Based on lines 132-133, it sounds like your approach should adapt to changes in channels, but I cannot see how that is expressed in the math/model.

2. In real-world scenarios, if the recording device is re-inserted across sessions, a different subset of neurons may be captured, which is a common challenge in neuroscience. Can your model be extended or generalized to cases where a potentially non-overlapping subset of neurons is observed across sessions (e.g., as done in [1])?  (I don’t expect the authors to change the model in this paper's scope, but it might be worth discussing in the discussion section)

3. It is not clear to me from line 109 if there is overlap between windows, or what you mean by “one step later” (line 110). Do you mean 1 sample after, based on the sampling rate? And does the overlap between windows depend on the sampling rate, or how else do you define the “step”?

4. Why is $ \tau $ restricted to the 0-1 range? Is it just due to the normalization of each trial to this range, which makes it easier to handle varying durations? I would suggest adding a short explanation of that in line 148.

5. How is $ \eta $ defined? How sensitive is the model to different choices of $ \eta $?

6. For the model to work, do the neurons need to be recorded simultaneously within each session?

7. Would the model work for brain areas (e.g., hippocampus) or animals (e.g., bats) with very sparse firing patterns, whose firing rates may not be well approximated by normal statistics? Would the model work for Poisson statistics data, which is more realistic for certain neural firing patterns?

8. Where is Monkey C in figure 3?

[1] Mudrik, N., Ly, R., Ruebel, O., & Charles, A. S. CREIMBO: Cross-Regional Ensemble Interactions in Multi-view Brain Observations. In The Thirteenth International Conference on Learning Representations.

**Relation To Broader Scientific Literature:**

The paper addresses an important challenge in modeling neural dynamics across sessions with limited trial data available in each session. The authors explain how their method addresses this issue and compare its performance to several benchmark approaches using rich monkey datasets.

**Theoretical Claims:**

Yes.

---

> ### Author Rebuttal · Authors · 2025-04-01
>
> We sincerely appreciate your thorough review and recognition of our work. Below we provide a detailed response to your concerns, with reference figures in the supplementary material [https://drive.google.com/file/d/129vv370SF4RLanLj92-lzh_vMmkeDCve/view?pli=1], indicated by Fig.R.
>
> ### Weaknesses
> - Model interpretability
>
>  The latent features effectively capture the evolved state within neural dynamics. We employ a transformer-based network $f_\alpha$ to extract latent features from signal windows. Given the challenges in interpreting the meaning of these features for realistic scenarios, our current focus is on the effectiveness of FDA for few-trial adaptation. We demonstrate the successful extraction of the Lorenz attractor from synthetic spiking data using our method below. We will explore the relationship between the latent features and underlying neural dynamics across various cases in future.
>
> - Synthetic data
>
> We conducted experiments to evaluate the recovery of ground-truth latent variables in synthetic data. Following method in (Kapoor, Jaivardhan, et al. Latent diffusion for neural spiking data. NeurIPS 2024: 118119), we used the Lorenz attractor as the latent dynamics. We simulated firing rates as an affine transformation of the 3D latent variables into a 96-dimensional space, then sampled spike trains from a Poisson distribution.
>
> As shown in the table, our FDA successfully recovered the latent dynamics from synthetic spiking data. The visualizations of our decoded 3D trajectories in Fig.R1 confirmed that FDA effectively captured the neural dynamics.
>
> #### Average $R^2$ (%) on the recovered latent variables
>
> | Mean Firing Rates | 0.05 | 0.1 | 0.3 |
> |:------:|:---------------:|:---------------:|:---------------:|
> | $R^2$    | 95.43 ± 0.87    | 95.68 ± 1.07    | 95.24 ± 1.03    |
>
> - About z (0-1)
>
> The evolution of z only represents the iterative learning process. The temporal evolution of neural dynamics is characterized by the shift of short-term windows(as explained in Line 106-108 on Page 2).
>
> ### Comments:
>
> - We will clarify the multi-dimensional labels in the revision.
>
> - Here $\tau$ represents the iterative learning steps instead of the temporal evolution of neural signals. To avoid confusion, we therefore used a distinct notation $\tau$ .
>
> - The scale coefficient represents the shift parameter of z generated by the MLP for predicting the velocity field.
>
> - Yes, it is all $l_2$ distance and will be standardized.
>
> - We find that our flow model can be well-suited to both few-trial adaptation in BCIs and non-BCI recordings. The flow process can include feedback as conditional features to guide the flow of subsequent windows.
>
> ### Questions:
> - Session definition
>
> The sessions used in experiments were from the same subject. The recorded neurons across sessions largely originate from the same subset. Moreover, as shown in Fig.R2, the transformer-based $f_{\alpha}$ is adaptable to varying numbers of tokens, corresponding to different channel counts.
>
> - Non-overlapping subset
>
> Given the widespread application of transformers for universal representation from neurons across regions, we believe our model has the potential to be extended to cases with non-overlapping neuron subsets across sessions. We agree that this is an interesting direction for future study.
>
> - One step later
>
> Yes, "one step later" refers to the next sample based on the sampling rate. The overlap between windows depends on the sampling rate, ensuring consistent time steps between the decoded and ground-truth variables.
>
> - $\tau$ (0-1)
>
> As explained above, $\tau$ represents the iterative learning step of z to obatin latent features, differing from the temporal evolution of neural signals.
>
> - Choice of $\eta$
>
> $\eta$ is pre-initialized using Xavier initialization. We tested various choices via differnt random seeds, and observed no significant impact on the final results (as shown in Table 1 on Page 7).
>
> - Simultaneous neurons
>
> The current model directly utilized temporal structures from single-channel recordings as input tokens for conditional features. Therefore, it is preferable for neurons to be recorded simultaneously within each session.
>
> - Poisson statistics
>
> As shown in the table above, the model demonstrated adaptability to Poisson synthetic data with various firing rates. Furthermore, as illustrated by the coefficient of variation (CV) distribution in Fig.R3, some real data we used exhibits characteristics of Poisson distributions.
>
> - Monkey C
>
> The figure for Monkey C is shown in Fig.S1(a).
>
> We sincerely hope that these responses may address your concerns. We believe that our novel FDA framework will be of significant interest to the ICML community, given its potential impact on few-trial neural alignment and real-world BCI reliability. Could you please consider raising the scores? We look forward to your further feedback. Thank you in advance.

---

> > ### Comment · Reviewer_3LVu · 2025-04-04
> >
> > I thank the authors for their response. I believe that the additional Synthetic data experiment is helpful.

---

### Decision · Program_Chairs · 2025-05-01

**Decision:**

Accept (poster)

**Comment:**

In work on brain computer interfaces (BCI), drift of recordings across days or sessions presents a difficulty for fixed decoders. If a decoder is trained on data collected from one neural population at a particular time, a change in the recorded population may drastically reduce decoder performance. Thus, "aligning" data across sessions or subjects in order to preserve the performance of an existing decoder is an important problem at the intersection of ML and neural engineering.

This paper presents a new approach to the neural alignment problem based on recent work in flow-matching. Briefly, an encoder is trained to compress spiking neural data into a context variable $\mathbf{c}$ that can be used to condition a latent flow $\mathbf{z}(t)$ from an isotropic Gaussian to a specified representation for decoding based on provided labels. The strategy of the paper is to retrain the encoder and flow on a large dataset, then fine-tune the encoder producing the flow-conditioning variable on a new few trials in a manner that minimizes a KL divergence between source and target decoding distributions.

Reviewers agreed that the proposed method is novel and addresses an important problem in the literature. They likewise agreed that the results on fine-tuning appear impressive, particularly with so few trials. Some questions were raised about the performance of some other large-scale models (e.g., POYO) with respect to fine-tuning, which will hopefully be clarified in a full revision.

In summary, this is a solid paper that shows real promise in addressing distribution shift for BCI applications.